# WHAT MATTERS IN LEARNING FROM LARGE-SCALE DATASETS FOR ROBOT MANIPULATION

**Vaibhav Saxena**[1], **Matthew Bronars**[1*], **Nadun Ranawaka Arachchige**[1*], **Kuancheng Wang**[1], **Woo Chul Shin**[1], **Soroush Nasiriany**[2], **Ajay Mandlekar**[3†], **Danfei Xu**[1,3†]
[1]Georgia Institute of Technology
[2]The University of Texas at Austin
[3]NVIDIA

## ABSTRACT

Imitation learning from large multi-task demonstration datasets has emerged as a promising path for building generally-capable robots. As a result, 1000s of hours have been spent on building such large-scale datasets around the globe. Despite the continuous growth of such efforts, we still lack a systematic understanding of what data should be collected to improve the utility of a robotics dataset and facilitate downstream policy learning. In this work, we conduct a large-scale *dataset composition study* to answer this question. We develop a data generation framework to procedurally emulate common sources of diversity in existing datasets (such as sensor placements and object types and arrangements), and use it to generate large-scale robot datasets with controlled compositions, enabling a suite of dataset composition studies that would be prohibitively expensive in the real world. We focus on two practical settings: (1) what types of diversity should be emphasized when future researchers *collect* large-scale datasets for robotics, and (2) how should current practitioners *retrieve* relevant demonstrations from existing datasets to maximize downstream policy performance on tasks of interest. Our study yields several critical insights – for example, we find that camera poses and spatial arrangements are crucial dimensions for both diversity in collection and alignment in retrieval. In real-world robot learning settings, we find that not only do our insights from simulation carry over, but our retrieval strategies on existing datasets such as DROID allow us to consistently outperform existing training strategies by up to 70%. More results at `https://robo-mimiclabs.github.io/`

## 1 INTRODUCTION

Imitation learning from offline datasets has emerged as a promising method to teach robots complex real-world manipulation tasks. Importantly, prior works have found that robot performance scales favorably with the dataset size and quality (Mandlekar et al., 2021; Brohan et al., 2022). Consequently, much efforts have been invested in building large-scale robot datasets that cover diverse tasks and environments. Recent large-scale efforts such as DROID (Khazatsky et al., 2024) and the Open X Embodiment datasets (Collaboration et al., 2023) have amassed millions of trajectories for table-top manipulation tasks. These datasets, while still orders of magnitude smaller than their vision and language counterparts, can allow robots trained on this data to generalize to different scenarios.

Despite the continuous growth and promising results of these data collection efforts, we still lack a systematic understanding of *what data should be collected* to improve the utility of a robotics dataset. However, gaining this understanding poses significant challenges. Data collection is extremely costly and time-consuming, often requiring teams of human operators, fleets of robots, and months of manual effort (Brohan et al., 2022), and the cost is compounded by the time and effort it takes to evaluate different robot models trained on these datasets. This makes testing the effectiveness of

---

*Equal contribution, †equal advising
Correspondence to `vsaxena33@gatech.edu`

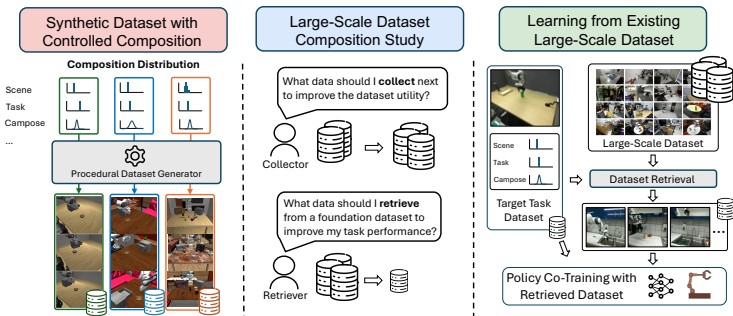

Figure 1: **MimicLabs overview.** Our framework encompasses: (1) A procedural dataset generator that creates diverse datasets with controlled composition. (2) A large-scale dataset composition study to analyze the impact of dataset diversity and alignment inspired by practical settings. (3) Extensive real-world experiments using existing large-scale robot datasets, based on study insights.

different dataset compositions intractable. The challenge is exacerbated by the sheer number of potential variations in dataset composition (e.g., object types, camera angles, lighting conditions), making it impractical to ask counterfactual questions about dataset composition ("what if the sorting task is collected with a mug instead of a plush toy"). As a result, current data collection efforts often rely on intuition rather than systematic analysis, potentially leading to inefficient use of resources.

To this end, we propose to use simulation and synthetic data generation as a testbed to answer critical questions about the collection and use of large scale datasets We develop a synthetic data generator that can procedurally emulate common sources of diversity found in existing datasets, allowing us to create large-scale robot datasets with precisely controlled composition. Leveraging this capability, we conduct a suite of dataset composition studies from multiple practical perspectives, aiming to provide insights into the efficient construction and utilization of large-scale robot datasets in the real world. Our work makes the following major contributions, as illustrated in Fig.1.

**1. Synthetic data generator for controllable dataset composition.** We develop a data generator to procedurally emulate common sources of diversity found in existing datasets, including sensor placements, object types and textures, and spatial arrangements. Our framework can synthesize diverse tasks and corresponding demonstration data by using a small set of human demonstrations, making it possible to generate large-scale datasets with controlled composition and enabling us to conduct a suite of dataset composition studies that would be prohibitively expensive in the real world.

**2. Practitioner-inspired collector and retriever settings.** We conduct our analysis from two practical perspectives. The *collector* perspective explores which dimensions of variation (DVs) should be prioritized when building large-scale datasets, focusing on the utility of increasing dataset diversity on broad skill transfer. The *retriever* perspective examines how to best utilize existing datasets for a specific target task, addressing questions such as whether to retrieve only the most similar demonstrations or train on the dataset in its full diversity. By connecting these two perspectives, we provide actionable insights on efficient dataset construction and utilization.

**3. Dataset composition studies with MimicLabs.** Leveraging our synthetic data generator, we create the MimicLabs dataset, a large-scale dataset comprising nearly 1M trajectories across over 3K task instances in 8 visually distinct simulation environments. This dataset reflects a realistic scenario where multiple robotics labs collaborate to collect diverse datasets. Our experiments with MimicLabs yield several key insights. For example, we find that camera poses and spatial arrangements are crucial dimensions for both diversity in collection and alignment in retrieval. High diversity in these dimensions enables better generalization, while alignment significantly boosts performance on target tasks. Conversely, we discover that object textures have minimal impact on downstream performance, suggesting that this dimension need not be prioritized in data collection or retrieval strategies.

**4. Insights that transfer to real-world settings**. Our study resulted in actionable insights readily applicable to real-world settings. We validate our findings on 7 real-world manipulation tasks, where both collector and retriever experiments corroborate our simulation-based insights. Particularly noteworthy are our retriever insights, which we applied to DROID (Khazatsky et al., 2024), an existing large-scale dataset. Our retrieval strategy, informed by the simulation studies, outperforms the existing approach of learning from the entire DROID dataset by up to 70% on certain tasks.

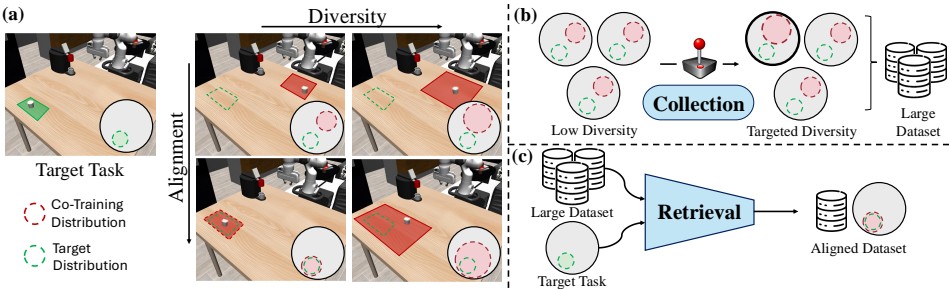

Figure 2: **Characterizing Co-Training Settings:** (a) We consider four co-training settings characterized by the diversity of their co-training datasets and their alignment with the target dataset. This is illustrated with the object spatial DV of the coffee pod. (b) The collectors aim to select and increase the diversity of selected DV(s) to improve the utility of a dataset. (c) The retrievers aim to extract a subset from an existing large dataset by aligning specific DVs with their target task in order to improve the target task performance.

## 2 RELATED WORK

**Large-scale Robot Manipulation Datasets.** There have been a number of recent efforts on collecting large-scale robot manipulation datasets. These include real-world datasets covering a broad set of manipulation tasks (Jang et al., 2022; Brohan et al., 2022; Mandlekar et al., 2018; Lynch et al., 2022; Walke et al., 2023; Bharadhwaj et al., 2023; Fang et al., 2023; Khazatsky et al., 2024)). Similarly, there have been efforts in collecting (Liu et al., 2023a) and generating (Mandlekar et al., 2023; Dalal et al., 2023; Nasiriany et al., 2024) data in simulation. In this work, we seek to improve future data collection and curation efforts by studying how dataset composition and different sources of diversity impact downstream policy learning.

**Study of Dataset Construction and Composition.** Recent works examine the role that data quality and curation play in policy generalization. Using data curation strategies to selectively weight data based on actions (Belkhale et al., 2023), interventions (Liu et al., 2023b), and datasets (Hejna et al., 2024) can yield benefits without collecting additional robot data. Other works explicitly focus on the dimensions of variation present in robot manipulation and study the role that these dimensions play in policy performance. Notably, Xie et al. (2023) and Pumacay et al. (2024) study policy performance under distributions shifts across different under fixed data distributions. Gao et al. (2024) compares policy performance under different data collection strategies with combinatorial variations. While these works separately study data curation and policy evaluation across different dimensions of variation, we bring a unified perspective of studying both questions and apply the resulting insights to improve existing practices in learning from a large-scale dataset (Khazatsky et al., 2024).

## 3 MIMICLABS STUDY DESIGN

A common use for large-scale robot datasets is for a practitioner to use them to boost the performance of a specific task that they would like their robot to perform. The goal of the **MimicLabs study** is to understand how dataset composition affects this downstream task performance. We first formalize the problem of co-training with large-scale multi-task datasets in Sec. 3.1. We then introduce a formalism for dataset composition in Sec. 3.2 and describe how this allows us to compare multiple datasets. Finally, Sec. 3.3 discusses how our experiments and analysis can help both future dataset collectors and current robotics practitioners for making the most out of large-scale robotics datasets.

### 3.1 PROBLEM FORMULATION

We study a practical scenario where a robotics practitioner trains a robot to perform a specific task of interest by leveraging a pre-existing, large-scale robot dataset (collected across diverse scenarios) and a small set of task-specific demonstrations. We refer to the specific task of interest as the **target task**. Formally, we denote the **target dataset**, which is the set of target task trajectories collected by the practitioner as $\hat{\mathcal{D}}_T = \{\xi_T^{(i)}\}_{i=1}^{N_T}$, which are samples from a demonstration distribution

$\mathcal{D}_T$. Here, $\xi_T^{(i)}$ represents a demonstration trajectory comprised of observation-action pairs. Each trajectory exhibits a certain behavior that carries out the target task in a well-defined environment setup. Similarly, we denote the **co-training dataset**, which is the large collection of available demonstrations as $\hat{\mathcal{D}}_C = \{\xi_C^{(i)}\}_{i=1}^{N_C}$, which are samples from a distribution $\mathcal{D}_C$. The practitioner then trains a visuomotor policy $\pi_\theta$ that learns to act by optimizing for $\theta$ as $\theta^* = \arg\min_\theta \mathcal{L}(\hat{\mathcal{D}}_T \cup \hat{\mathcal{D}}_C; \theta)$, where $\mathcal{L}$ is a behavior cloning (BC) objective. Following prior works (Fu et al., 2024; Khazatsky et al., 2024), we refer to this strategy of learning from the combined dataset as "co-training."

### 3.2 Dimensions of Variation (DVs) and Co-Training Settings

To characterize dataset composition, we need to formalize a demonstration distribution $\mathcal{D}$. We introduce the concept of **Dimensions of Variations (DVs)**, which are independent factors that characterize the variability in demonstration data. We denote the full set of distributions that create variation in $\mathcal{D}$ as $\{\mathcal{Z}^{(1)}, \ldots, \mathcal{Z}^{(K)}, \tau\}$, where $\mathcal{Z}^{(k)}$ is the distribution of variation along a single DV in the environment, and $\tau$ denotes the goal of the task. We consider $\mathcal{D}$ to be parameterized as $\mathcal{D}(\mathcal{Z}^{(1)}, \ldots, \mathcal{Z}^{(K)}, \tau)$. An important criterion for DV selection is that each DV can be measured in each demonstration of a large heterogeneous dataset such as DROID (Khazatsky et al., 2024). This allows us to study different dataset compositions on existing robotics datasets by retrieving subsets of the dataset that might be measurably diverse or aligned with the target distribution. For example, we study if retrieving demonstrations from DROID that are aligned along the camera pose DV helps boost downstream performance, where we use the camera extrinsics as the DV measurement. Finally, we note that our analysis framework and tools are not tied to the specific set of DVs and can be utilized for any additional DVs of interest.

Given parameterized dataset with DV distributions, we can characterize the relationships between target ($\mathcal{D}_T$) and co-training ($\mathcal{D}_C$) demonstration distributions. In particular, we aim to understand how the **diversity** of $\mathcal{D}_C$ and the **alignment** between $\mathcal{D}_C$ and $\mathcal{D}_T$ impact co-training performances. We define $\mathcal{S}(\cdot)$ as the support of any distribution and $|\mathcal{S}(\cdot)|$ as a measure of its size. We say $\mathcal{D}$ is **diverse** along DV $k$ if $|\mathcal{S}(\mathcal{Z}^{(k)})|$ is large. Additionally, we say that $\mathcal{D}_T$ and $\mathcal{D}_C$ are **aligned** along $\mathcal{Z}^{(k)}$ if $\mathcal{S}(\mathcal{Z}_T^{(k)}) \subset \mathcal{S}(\mathcal{Z}_C^{(k)})$. This creates four cases when comparing $\mathcal{D}_T$ and $\mathcal{D}_C$ along a DV, as illustrated in Fig. 2 with more details in Appendix B. In our experiments, considering these cases help us understand which DVs need to be diverse in the co-training dataset, and how important alignment is for target task performance.

### 3.3 Practical Settings: The Collector and the Retriever

We seek to understand the effects of dataset composition through two different lenses inspired by practical scenarios: a data **collector** and a data **retriever**.

**Data Collector.** We consider **collectors** as practitioners who are contributing demonstrations to a large robotics dataset. Since data collection is time-consuming, we seek to understand which DVs should be prioritized for diversity or alignment during data collection to facilitate downstream task learning. Instead, in our experiments, we set $\mathcal{D}_T$ and $\mathcal{D}_C$ to be fully aligned along all DVs except one. This allows us to study the impact of a single DV (say $k$) in $\mathcal{D}_C$, by constructing different co-training datasets $\mathcal{D}_C^{(k)}$ where each dataset corresponds to a different choice of distribution for DV $k$, $\mathcal{Z}_C^{(k)}$. For example, to study the impact of different camera poses, we would construct $\mathcal{D}_C$ to be aligned with $\mathcal{D}_T$ along spatial arrangements, textures, background, etc. and just create datasets with different camera pose distributions ($\mathcal{Z}_C^{(k)}$).

**Data Retriever.** A data **retriever** is a practitioner that seeks to use a pre-existing large-scale co-training dataset $\hat{\mathcal{D}}_C$ with high variation in several DVs (such as DROID) for their specific task of interest. In this setting, we aim to retrieve a subset $\hat{\mathcal{D}}_{C'} \subset \hat{\mathcal{D}}_C$ to maximize downstream task performance, by aligning some DVs from the retrieved co-training dataset $\mathcal{Z}_{C'}^{(k)}$ with those from the target dataset $\mathcal{Z}_T^{(k)}$. For example, we study whether retrieving demonstrations from DROID that have aligned camera poses with that of the downstream setup will improve learning performance. In certain cases it can be difficult or impossible to achieve such alignment – consequently, our experiment also studies the effect of retaining diversity in DVs (e.g. a DV $\mathcal{Z}_{C'}^{(k)}$ with large support).

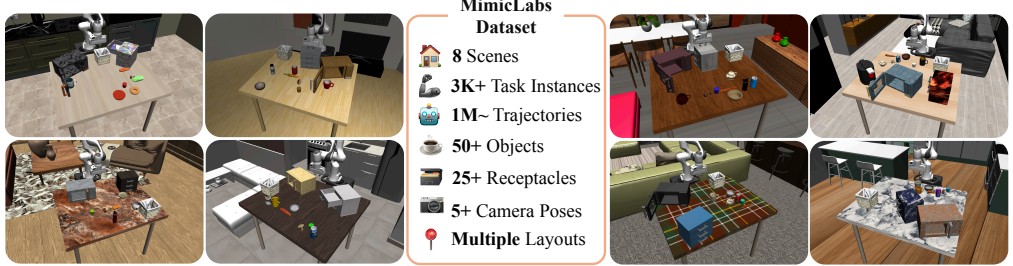

Figure 3: **MimicLabs Dataset.** We use our data generator to create a large-scale, multi-task dataset to emulate a realistic scenario where multiple labs collaborate to create a dataset with large variations in diverse DVs. We leverage this dataset to conduct our dataset composition study.

## 4 PROCEDURAL TASK AND DEMONSTRATION GENERATION

To understand how different data composition choices influence downstream policy learning, we develop a framework for procedural task and demonstration generation. This framework enables us to generate large multi-task datasets with controlled composition, culminating in the creation of our MimicLabs dataset (Fig. 3). Our framework consists of two main components: procedural task generation and procedural demonstration generation. It takes a set of dataset composition factors as input (DVs, Sec. 3.2) and uses them to generate a diverse set of task instances, followed by demonstration generation for each instance.

**Procedural task specification.** To enable controllable data generation for large multi-task datasets, we require a way to specify tasks that allows for diverse procedural generation. These task instances should vary along specific Dimensions of Variation (DVs, Sec. 3.2), which allow us to parameterize the composition of these datasets. To this end, we leverage the Behavior Domain Definition Language (BDDL) (Ghallab et al., 1998; Srivastava et al., 2021). The BDDL grammar allows for specifying scenes, objects, their spatial arrangements, and predicates for task initialization and success. We build upon the parsing framework of LIBERO (Liu et al., 2023a) and generate task specifications with controllable variation in spatial arrangements of objects and receptacles, camera poses, and textures. These are combinatorially varied, resulting in a large set of task instances for collecting and generating demonstrations for our study. Details about the task specification are in Appendix D.

**Procedural demonstration generation.** To automate demonstration generation on our procedurally generated task instances, we build upon MimicGen (Mandlekar et al., 2023). Unlike prior studies that use "scripted" experts (Xie et al., 2023) or rely entirely on human teleoperation (Mandlekar et al., 2021; Liu et al., 2023a), our approach allows for scalable data generation while preserving the fine-grained manipulation strategies used in human demonstrations. MimicGen automates data generation by decomposing source human demonstrations into object-centric manipulation segments, which are then transformed and stitched together to create new trajectories for novel scenes. Our key innovation lies in leveraging the BDDL task specification and the state predicates provided by the simulator to largely automate the process of decomposing tasks into these segments. This automation significantly reduces the human effort required when scaling to thousands of task instances. By using MimicGen in conjunction with our BDDL-based task specifications, we can scale from a small set of human demonstrations to a large, diverse dataset. This approach allows us to generate demonstrations across wide variations in DVs from a limited set of source demonstrations, and easily generate demonstrations for new tasks by swapping out the BDDL task specification. For example, given source demonstrations of a robot picking up a bowl with a fixed texture and putting it in a basket from a shoulder camera view, our pipeline can create datasets with varying camera poses, bowl textures and geometries, table textures, and spatial arrangements – all specified in BDDL.

**The MimicLabs Dataset.** We can now use our controllable multi-task dataset generation framework to collect large-scale multi-task datasets by conditioning on different dataset composition choices (DVs, Sec. 3.2). To fully leverage the power of our framework, we curate a large set of DV ranges that can be combinatorially varied, including camera poses, objects, receptacles, and their textures and spatial arrangements, and instantiate over 3000 task instances with language instructions in 8 different scenes. We then collect ∼500 source human demonstrations for a subset of these task instances and synthesize over 1M demonstrations using our procedural data generation framework. We refer to our

Table 1: **Analyzing the effects of misaligned and diversities in DVs from a collector's perspective.** Details about baseline variation are in Appendix F.2. Target variations are perturbations to one DV distribution while others stay fixed. Success rates with co-training variations show how increasing variation in one DV may help skill transfer from co-training in that DV as well as other DVs.

| DV w/ target-cotrain misalignment | Target only | Co-training with different DV distributions | | | | |
|---|---|---|---|---|---|---|
| | | Baseline | camPose | objTex | tableTex | objSpat |
| camPose | 16.67 | 43.33 | **90** | 43.33 | 43.33 | 30 |
| objTex | 30 | **93.33** | 96.67 | 90 | 90 | 93.33 |
| tableTex | 43.33 | 66.67 | **80** | 63.33 | **83.33** | 70 |
| objSpat | 10 | 26.67 | 46.67 | 33.33 | 26.67 | **56.67** |

dataset as the **MimicLabs dataset**. This dataset is meant to reflect a realistic scenario where multiple robotics labs collaborate to collect datasets with large amounts of variation in diverse DVs, as prior work has done in real-world settings (Khazatsky et al., 2024). We illustrate the different scenes containing objects and receptacles in Fig 3. Our dataset consists of both long- and short-horizon tasks varied across multiple DVs that can be used to either learn specific skills, stitched together to train complex behavior, or used to evaluate new robot policies for their generalization capabilities. See Appendix C for more details of DV distributions for generating this dataset.

## 5 CORE RESULTS

We leverage our MimicLabs pipeline for demonstration generation and collect datasets of varying diversities and relative alignments. We use these datasets to gather takeaways from a data collector and retriever's standpoint, and finally validate our findings in both settings through experiments on a real robot. Training details for all experiments can be found in Appendix E.

### 5.1 EFFECT OF INDIVIDUAL DVS: COLLECTOR'S PERSPECTIVE

Data collectors are practically limited in the number of trajectories they can collect, hence it is important to understand where time is best spent when adding demonstration diversity. We investigate which DV should be diverse in the co-training dataset to maximize downstream task success. Our experiments are designed to determine when **alignment** is necessary for a certain DV, and when increasing **diversity** helps alleviate misalignment in composed datasets. To do this, we construct target-cotraining distribution pairs that are aligned along all but one DV, including a large baseline dataset with low variation and misalignment along that DV. By combining the target dataset with different co-training datasets (for each DV), we identify DVs where it is crucial to have diversity during data collection to mitigate misalignment along that and other DVs. This comprehensive approach helps collectors understand where to focus their efforts best when adding diversity to their datasets for optimal downstream performance.

**Experimental Setup.** We examine a *clear table* task where the robot must open a cabinet's top drawer and place a bowl inside, amidst four distractor objects. This task requires two steps: `openTopDrawer` and `pickPlaceTopDrawer`. We identify and independently vary five DVs: camera pose (*camPose*), object texture (*objTex*), table texture (*tableTex*), object spatial arrangement (*objSpat*), and motion primitive. Our experiments test how co-training diversity in these DVs affects transfer learning when one DV is misaligned with the target. Additionally, we investigate motion diversity and alignment effects on two task variants: (1) `pickPlaceTopDrawer+push`, where the robot places the bowl in an open drawer and closes it, and (2) `pull+pickPlaceTopDrawer`, where the robot opens a closed drawer before placing the bowl inside. For these variants, we maintain alignment across all other DVs. Results are in Tables 1 and 10, with our findings discussed below. Additional results on a *make coffee* task are in Table 9.

**Less diverse and misaligned camera poses prevent skill transfer.** We found that co-training with a dataset that has a misaligned camera pose significantly hurts the boost in performance the co-training dataset could have otherwise offered (in Table 1 for misalignment along camPose, notice low Baseline performance). This points to the fact that camera poses are *necessary to be aligned* between target and co-training distributions for maximal performance boost. In this setup, we find that increasing

Table 2: **Retriever's Perspective.** Task success when co-training using various retrieval strategies, including counterfactual cases (i.e., misaligned retrieval).

| Target task | Target only | Full variation | Camera pose | | | Object texture | | | Table texture | | |
|---|---|---|---|---|---|---|---|---|---|---|---|
| microwave mug | 6.67 | 60 | **70** | 46.67 | 53.33 | 66.67 | 73.33 | 70 | 70 | 40 | 43.33 |
| clear table | 13.33 | 30 | **36.67** | 30 | 20 | 33.33 | 26.67 | 30 | 36.67 | 23.33 | 40 |

| Target task | Object spatial | | Receptacle spatial | | |
|---|---|---|---|---|---|
| microwave mug | **70** | 56.67 | **46.67** | 16.67 | 20 |
| clear table | **40** | 26.67 | **30** | 23.33 | **30** |

the diversity of camera poses in the co-training dataset improves the transfer, even when the target camera pose distribution is mis-aligned with the co-training distribution. For the *clear-table* task, we see over $40\%$ performance boost (in Table 1 same row, see Baseline vs camPose).

**Co-training with misaligned object textures, with minimal variation, was sufficient for downstream success.** We find that simply training with a large set of co-training demos, without any heed to aligned or even varied object textures, led to successful task completion. Specifically, we see over $90\%$ success in the target task despite completely misaligned object textures with co-training. Note that object geometries were consistent between target and co-training, variation in which we analyze in a later section. Also note that we did not vary lighting conditions or material properties in the target tasks, leading to the rendering of the object primarily dependent on its texture.

**High diversity in camera poses assists transfer learning with misaligned textures.** We find that co-training datasets with high diversity in camera pose can transfer to target tasks with misaligned table textures and object textures. Changing camera poses allows the robot to see larger background variations, potentially leading to this visual robustness. This observation leads us to believe that adding *camera pose diversity is generally useful* during data collection.

**Spatial coverage in co-training is critical to downstream performance.** High diversity in spatial arrangement in the target task necessitates diverse action coverage for the robot, which is not covered by any visual DV. We find that adding diversity in the spatial arrangement of objects was the only way we could significantly boost transfer learning in the presence of misalignment along this DV (in Table 1 for misalignment in objSpat, see Baseline vs objSpat) While increasing diversity in camera poses adds some performance boost, the best-performing co-training is still the one with largest diversity in spatial arrangements. This leads us to believe that spatial arrangement is both *necessary to be aligned* and *generally useful for diverse data collection*.

**Co-training with aligned motion primitives and high diversity enables better downstream performance.** We find that when co-training was done with just a single motion primitive, it is significantly more important to make sure that the motion was aligned with that of the target, or else performance sometimes degraded (by 20% on the easy variant of *clear table*, see Table 10). However, this was of less importance as diversity in motion was increased as we saw a steady increase in success rate. For both variations of the *clear table* task, the co-training dataset with maximum coverage of the target *motion* provided the maximum boost to performance in the target task, leading us to believe that *robot motion is necessary to be aligned*.

## 5.2 Effect of Individual DVs: Retriever's Perspective

Now that we understand which DVs are crucial for transfer learning from a co-training dataset, we try to understand if these takeaways can help us build strategies for retrieving demonstrations from large robotics datasets for maximal downstream success (as described in Sec. 3.3). In this section, we design experiments to analyze each DV that can be independently retrieved from a large robotics dataset, and try to understand where it matters to have maximal alignment between target and co-training distributions. Additionally, we answer counterfactual questions covering scenarios where good retrieval is not possible, and if heavy variation in all DVs in co-training is the answer to boosting downstream performance. Overall, we answer the following two questions for retrieval: (1) when does it matter to have maximal alignment between target and co-training, and (2) when alignment is not possible, does having high diversity help?

Table 3: **MimicLabs Retrieval.** Task success in the MimicLabs benchmark when co-training using various retrieval strategies.Results in **bold** are best co-training within the sub-experiment, but only if performance boost $\geq 5\%$ than target only. We provide the number of demonstrations retrieved for each experiment in Table 8.

| Target task | #demos | Target only | Retrieving relevant object/skill | | | | | Missing relevant object/skill | | | | |
|---|---|---|---|---|---|---|---|---|---|---|---|---|
| | | | ✓ *obj/skill* | *+ camPose* | *+ objSpat* | *+ recepSpat* | *+ all* | ✗ *obj/skill* | *+ camPose* | *+ objSpat* | *+ recepSpat* | *+ all* |
| **bin carrot** | 10 | 50 | 70 | **96.67** | 86.67 | 83.33 | 90 | 30 | 40 | 53.33 | 43.33 | **56.67** |
| **bin bowl** | 10 | 33.33 | 50 | 70 | 53.33 | 63.33 | **73.33** | 36.67 | **60** | 50 | 40 | 46.67 |
| **clear table** | 10 | 23.33 | 20 | 20 | 20 | 20 | 23.33 | 20 | 23.33 | 20 | 20 | 16.67 |
| | 20 | 36.67 | 43.33 | **46.67** | 43.33 | 43.33 | 40 | 33.33 | 33.33 | 23.33 | **43.33** | 30 |
| **microwave teapot** | 10 | 23.33 | 20 | 23.33 | 16.67 | 13.33 | 16.67 | 10 | 10 | 6.67 | 13.33 | **20** |
| | 20 | 30 | 33.33 | **50** | 33.33 | 36.67 | 33.33 | 20 | **36.67** | 26.67 | 33.33 | 23.33 |
| **make coffee** | 10 | 13.33 | 10 | **23.33** | 6.67 | 13.33 | 6.67 | 3.33 | 13.33 | 6.67 | 6.67 | 6.67 |
| | 50 | 33.33 | 33.33 | 36.67 | 30 | 36.67 | 33.33 | 30 | 30 | 30 | 30 | **40** |
| *top-3 labs* | 50 | 33.33 | 30 | **53.33** | 40 | 40 | 40 | 36.67 | 36.67 | **43.33** | 30 | 40 |

**Experimental Setup.** We analyze two tasks: *microwave mug* and *clear table*. These tasks are challenging enough to benefit from co-training, as evidenced by low success rates with just 10 expert demos. For each DV, we create three co-training datasets: fully aligned with the target distribution, misaligned and low diversity, and misaligned with high diversity. We compare these against a baseline dataset with full combinatorial variation in all DVs, including the target distribution. All experiments use 10 target demos and 1000 co-training demos. We include highlight results in Table 2, with full results in Fig. 13.

**Camera pose alignment and diversity significantly impact performance.** Aligning camera poses in the retrieved dataset substantially boosts transfer learning, as it provides the robot with relevant examples for the target viewpoint. Our results show a large performance gap when camera poses were aligned with the target (shoulder-left/right) compared to when they were completely misaligned (agent-front). When perfect alignment is impossible, high diversity in camera poses can still improve performance by preventing overfitting to specific hand-eye coordinations. This suggests that data collectors should prioritize camera pose variation to enable effective retrieval for downstream tasks

**Object texture alignments have limited impact in retrieval.** We find that in our setup object texture did not matter at all for either alignment or diversity. That is to say that the target demonstrations were enough for the model to understand what texture it needs to attend to and hence is something of minimal importance to a data collector or a retriever.

**Spatial arrangement alignment and diversity are crucial for performance.** Aligning spatial arrangements between co-training and target datasets significantly boosts downstream performance for both objects and receptacles. This is because spatial variations directly impact the robot's action space coverage. When perfect alignment is not possible, high variation in spatial arrangements can still improve performance, especially when target arrangements aren't fully covered. Conversely, low variation in misaligned spatial arrangements can lead to overfitting and poor skill transfer.

## 5.3 RETRIEVER STUDY WITH MIMICLABS

Having built some understanding of which DVs matter for downstream success when collecting data and retrieving it in a pedagogical setup, we now use those takeaways to build strategies for retrieving data from large simulated robotics datasets. As described in Sec. 4, we created a dataset containing combinatorially varied object and spatial arrangements, camera poses, and tasks, in 8 different scenes, simulating a real-world setup where mutliple labs contribute to form a large composite dataset that each lab uses to boost downstream performance for a target task they care about. We consider 5 target tasks in the MimicLabs benchmark for this experiment (increasing order of hardness): *bin carrot*, *bin bowl*, *clear table*, *microwave teapot*, and *make coffee*. Details are in Appendix F.1.

**Retrieval Strategies.** We perform 2 kinds of retrievals on the MimicLabs dataset for each target task: (1) retrieving demos that contain the target object or skill (such as grasping a bowl, pulling a drawer), and (2) a counterfactual retrieval that ignores any demos that might contain them. The latter simulates a situation where a retriever may not find any skill that is useful for their target task but still might try to use this dataset in the hopes of some performance boost. In either case, we do a subsequent retrieval on camera poses and spatial arrangements to show performance boost that the retriever might gain by aligning along these DVs. We summarize these results in Table 3.

Figure 4: **Real World Tasks.** We leverage our insights from the simulation study to conduct experiments with 7 manipulation tasks in the real world.

**Retrieving skills for co-training, even in the presence of overall heterogeneity, significantly boosts downstream performance.** For every target task we considered, we find that retrieving and aligning *skills* with co-training had a significant impact on task performance. This difference was significantly large for the easier tasks with a single atomic subtask ($40\%$ boost for *bin carrot*) which shows that the model was able to significantly make use of the required skill when there was little heterogeneity in target motion. Moreover, this also tells us that diversity in object geometries during data collection, which allows for eventual skill retrieval, should be a useful DV for a data collector.

**Aligning camera poses enables better transfer of skills.** For almost all co-training experiments where skills were aligned between target and co-training, the dataset with aligned camera poses gave a further boost to co-training. It was also the best-performing co-training for *make coffee* with $20\%$ boost over target-only performance compared to just partial skill alignment (picking and placing coffee pod) which showed no performance boost.

**Quality often matters over quantity in co-training data.** Retrieving skills in our dataset usually resulted in $\frac{1}{10}$th the amount of data than when skills could not be retrieved ($\mathcal{O}(100k)$ demos to $\mathcal{O}(10k)$ demos). However, it almost always resulted in better downstream performance. Even more, a full retrieval along camera poses and object/receptacle spatial arrangements resulted in $\mathcal{O}(1k)$ demos, and still sometimes outperformed other strategies. This proves that visuomotor policies are inherently prone to unstable training in the presence of heterogeneous data, highlighting the importance of diversity in data collection and subsequent retrieval.

## 5.4 Real World Experiments

Finally, we seek to demonstrate that the findings from our simulation study transfer over to real-world settings. We replicate the collector and retriever experiments in the real world. Notably, we perform retriever experiments on an existing large-scale dataset (DROID) and compare full-dataset co-training as adopted in prior works with different co-training strategies inspired by our simulation study.

**Experimental Setup.** We design our hardware setup to match that of DROID (Khazatsky et al., 2024) – a Franka robotic arm with a Robotiq gripper, mounted on a mobile platform, with several externally mounted cameras (details in Appendix H). All real-world policies are trained with Diffusion Policy (Chi et al., 2023) and evaluated across 20 rollouts (details in Appendix E). We evaluated seven different tasks across the collector and retriever perspectives of analyses. One of these tasks - *store screwdriver* was only used for collector experiments. Two tasks - *bin can* and *baking* were used for both collector and retriever while four other tasks - *serve snack*, *pour*, *wipe board*, and *put marker in cup* were used only for retriever. The tasks were chosen to represent a variety of manipulation capabilities including high-precision manipulation, non-prehensile manipulation, and large spatial generalization. We designed our retriever tasks around objects with sufficient demos in DROID (e.g. DROID includes 4,500 task instances that involve markers). Full details are in Appendix F.3.

### 5.4.1 Collector's Perspective (Real World)

Similar to our collector experiments in simulation, we evaluate models that are co-trained on various distributions of a single misaligned DV, while the rest of the DVs are fully aligned. To validate our collector experiments in simulation, we focus on three DVs - *camPose* , *objSpat* , and *objTex* , and evaluate whether these DVs have a similar impact on real-world policy learning results. Results are presented in Table 13 in Appendix K across three different tasks. See Appendix I for task details.

**Main findings.** Our experiments reveal several key insights about co-training effects across different DVs. Camera pose alignment proves crucial, with performance improvements of 25-50% when the co-training dataset matches the target camera distribution. Interestingly, co-training with misaligned

textures can still enhance performance by 40-65%. This aligns with our simulation findings and suggests that models learn relevant motions and skills without necessarily focusing on exact object colors. Additionally, increasing the diversity of spatial distributions in both target and co-training datasets significantly boosts performance, as exemplified by the *bin can* task where performance improved from 30% with a small spatial distribution to 60% with a large one. This corroborates our simulation results, demonstrating that broader spatial coverage expands the action space, leading to improved task execution.

### 5.4.2 RETRIEVER'S PERSPECTIVE (REAL WORLD) USING DROID

To validate whether our study insights are broadly useful, we apply insights from the retrieval setting to the DROID dataset (Khazatsky et al., 2024), a large-scale dataset collected across several robotics labs, and use it for specific manipulation tasks on our robot setup.

**Retrieval from DROID.** To facilitate structured retrieval from DROID, we pre-processed the dataset by filtering demos with language instructions, labeling manipulated objects, annotating object colors using a VLM, and marking object positions based on gripper actions (details in Appendix J). We evaluated retrieval strategies aligning specific DVs, comparing them against models trained on target demos only and co-trained with the entire DROID dataset. Results are presented in Fig. 5.

Figure 5: **Retriever results on a real robot.** We compare the performance of models trained on retrieved datasets with those co-trained on the entirety of DROID.

| Target task | #demos | Target only | DROID | *obj/skill* | *+camPose* | *+objTex* | *+objSpat* |
|---|---|---|---|---|---|---|---|
| *serve snack* | 20 | 5 | 0 | 65 | 70 | 35 | **85** |
| *bin can* | 20 | 60 | 0 | 15 | **85** | 65 | **85** |
| *pour* | 20 | 50 | 0 | 35 | **75** | 60 | 65 |
| *wipe board* | 20 | 55 | 0 | 45 | 55 | 55 | **65** |
| *baking* | 20 | 40 | 0 | 40 | **55** | 40 | 35 |
| *put marker in cup* | 50 | 30 | 0 | 20 | **35** | 15 | 20 |

**Aligned retrieval can significantly improve performance.** For all of our evaluated tasks, we see that co-training with retrieval outperforms models trained on just target demos, ranging from a 5% to 80% boost in performance. The highest increases in performance always came from retrievals that aligned camera poses or spatial locations of the target objects. For instance, for the *pour* task, we see a 25% increase in performance when co-training with aligned camera poses, with the success rate decreasing by 15% when retrieving only the relevant object without aligning camera poses or spatial locations. Qualitatively, we found that models co-trained with aligned retrieval had more robust retrying behavior and better precision.

**Random co-training with a large dataset can hurt performance.** All of the models we co-trained with all of DROID failed to learn the tasks, showing that too much diversity in co-training datasets can potentially lead to unstable training and diminished performance. Qualitatively, we find that models co-trained on all of DROID had erratic movement and were not precise enough to complete the target tasks. We believe this happens as co-training on a large dataset might cause a model to learn motions and features that are unnecessary for completing the target task.

**Retrieval is a promising direction and standardization of dataset metadata will be beneficial.** We had to design and create our own metadata in addition to what was found in DROID, which is a non-trivial effort. When collecting large numbers of robot demonstrations, it is probably beneficial to include detailed task descriptions and metadata, and a way to query this, to facilitate easy and accurate retrieval for future robot learning researchers.

## 6 CONCLUSION

Our study offers valuable insights for both collectors and users of large-scale robotics datasets. For data collectors, we highlight the importance of prioritizing diversity in camera poses and spatial arrangements, while suggesting that extensive variation in object textures may be less critical. For practitioners, we demonstrate the benefits of strategic data retrieval, showing that aligning critical dimensions between co-training and target task distributions can significantly boost performance, often outperforming training on entire diverse datasets. These findings can inform more effective data collection and utilization strategies in the community.

## 7 ACKNOWLEDGMENT

This work was supported in part by the National Science Foundation Award 2409016 and by a gift from Autodesk. The opinions, findings, and conclusions or recommendations expressed are those of the authors and do not necessarily reflect the views of the National Science Foundation or Autodesk.

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

# A OVERVIEW

The Appendix contains the following content.

- **Defining the four cases of dataset distributions** (Appendix B): this section describes the 4 different cases of target and co-training distributions considered in this study.

- **Description of each dimension of variation (DV) in MimicLabs** (Appendix C): this section describes the different dimensions of variation we consider in our combinatorial data generation pipeline as well as for our study.

- **Task Specification via BDDL** (Appendix D): this section specifies details about the parameterization for different procedurally generated DVs in MimicLabs.

- **Training and Evaluation Details** (Appendix E): this section lists out the policy hyperparameters for BC-RNN and Diffusion Policy, as well as other co-training details.

- **Task Descriptions** (Appendix F): this section provides descriptions of different target tasks we experimented on in MimicLabs and on a real robot.

- **Details about Retriever Experiments** (Appendix G): this section specifies the number of demos used for each retriever experiment in MimicLabs.

- **Real Experiment Setup** (Appendix H): this section describes our hardware and setup for the real-world experiments.

- **Real Collector Datasets** (Appendix I): this section elaborates on the data collection for the real-world collector experiments.

- **DROID Retrieval and Metadata Processing** (Appendix J): describes how retrieval was performed on DROID for the different DVs.

- **Additional Results** (Appendix K): this section provides additional results.

## B DEFINING THE FOUR CASES OF DATASET DISTRIBUTIONS

Given target and co-training demonstration distributions $\mathcal{D}_T$ and $\mathcal{D}_C$, $\mathcal{S}(\cdot)$ representing the support of a distribution and $|\mathcal{S}(\cdot)|$ the measure of its size, we define four cases comparing the supports of these distributions and sizes:

1. not-diverse and misaligned: $|\mathcal{S}(\mathcal{Z}_C^{(k)})| \approx |\mathcal{S}(\mathcal{Z}_T^{(k)})|$ and $\mathcal{S}(\mathcal{Z}_T^{(k)}) \cap \mathcal{S}(\mathcal{Z}_C^{(k)}) = \phi$

2. diverse and misaligned: $|\mathcal{S}(\mathcal{Z}_C^{(k)})| \gg |\mathcal{S}(\mathcal{Z}_T^{(k)})|$ and $\mathcal{S}(\mathcal{Z}_T^{(k)}) \cap \mathcal{S}(\mathcal{Z}_C^{(k)}) = \phi$

3. diverse and aligned: $|\mathcal{S}(\mathcal{Z}_C^{(k)})| \gg |\mathcal{S}(\mathcal{Z}_T^{(k)})|$ and $\mathcal{S}(\mathcal{Z}_T^{(k)}) \subset \mathcal{S}(\mathcal{Z}_C^{(k)})$

4. not-diverse and aligned (perfect alignment): $|\mathcal{S}(\mathcal{Z}_C^{(k)})| \approx |\mathcal{S}(\mathcal{Z}_T^{(k)})|$ and $\mathcal{S}(\mathcal{Z}_T^{(k)}) \subset \mathcal{S}(\mathcal{Z}_C^{(k)})$

We illustrate these cases comparing $\mathcal{D}_T$ and $\mathcal{D}_C$ along a DV in Fig. 2, 1-4 starting from top-left going in clockwise order. As shown in our experiments, cases 1 and 2 (diverse or not, with misalignment) are important cases to study from a collector's perspective where the goal is to find out which DV the collector should add diversity in given the worst-possible scenario i.e. misalignment. Cases 3 and 4 are important from a retriever's perspective given the assumption that large-scale robotics datasets contain high diversity along all DVs. Subsequently, our study seeks to understand if retrieving datasets with perfect alignment between target and co-training datasets is essential for boosting performance in the target domain.

## C DESCRIPTION OF EACH DIMENSION OF VARIATION (DV) IN MIMICLABS

*Camera Pose (camPose)*: in the MimicLabs dataset we vary the camera pose anchored at the center of the table and also such that it always points towards the center of the table while its position is varied in spherical coordinates (physics convention). Camera positions are grouped into 5 possible bins: shoulder-left and shoulder-right w.r.t the robot, and agent-left, agent-right, agent-front w.r.t. an agent. Each bin in the MimicLabs dataset is comprised of a variation of 15 degrees in the polar angle and 30 degrees in the azimuthal angle.

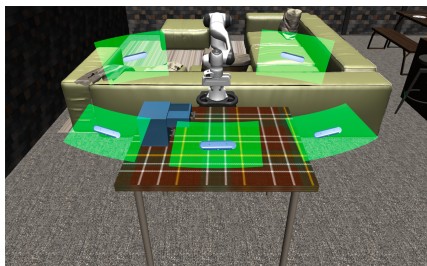

Figure 6: Illustrating camera distributions in MimicLabs discretized into 5 bins.

*Object Texture (objTex)*: Each task has a target object and we vary the object color / texture procedurally using generated fractal noise.

*Table Texture (tableTex)*: We vary the color/texture of the table surface procedurally using fractal noise.

*Target Object Spatial Arrangement (objSpatial)*: The placement of the target object is randomly initialized on the table and we vary the size of this reset range.

*Receptacle Spatial Arrangement (recepSpatial)*: For tasks that involve placement of an object into a receptacle, we randomly initialize the location of the receptacle. We vary the size of this reset range.

*Background Scene (scene)*: Simulation tasks take place in 1 of 8 visually distinct lab environments. Target tasks for real world experiments take place in one lab is co-trained on DROID (Khazatsky et al., 2024) data from 52 buildings.

*Motion Primitives (motion)*: Different tasks are composed of different motions - we segment these as `pick`, `place`, `push` and `pull` in the data composition experiments from the collector's perspective.

# D  TASK SPECIFICATION VIA BDDL

Our task specification in the MimicLabs dataset follows the following parameterization for sampling multiple DVs at initialization:

- *spatial arrangements:* specified using a union of multiple bounding boxes on a table-top. These bounding boxes are represented as a 4-tuple specifying the start and end (x,y) locations in the robot's base frame.

- *camera poses:* specified using a union of ranges in spherical coordinates (physics convention) in a coordinate frame with the origin at the center of the table. Each range is represented as a 6-tuple specifying the $(r, \theta, \phi)$ (i.e. radial, polar, azimuthal) ranges for the camera placement in a frame anchored at the center of the table.

- *textures:* specified as HSV ranges to either jitter a base texture retrieved from a file into the specified range (used for table), or fully generated as a fractal noise within the specified HSV range (used for objects).

- *demonstration:* specified as predicates that determine what motion primitives, and in what order, are useful for task completion. These predicates also split the task in an object-centric fashion that allows for large-scale reuse of human demonstrations across a variety of tasks that follow same primitives (using MimicGen (Mandlekar et al., 2023)).

## D.1  TEXTURES IN THE MIMICLABS DATASET

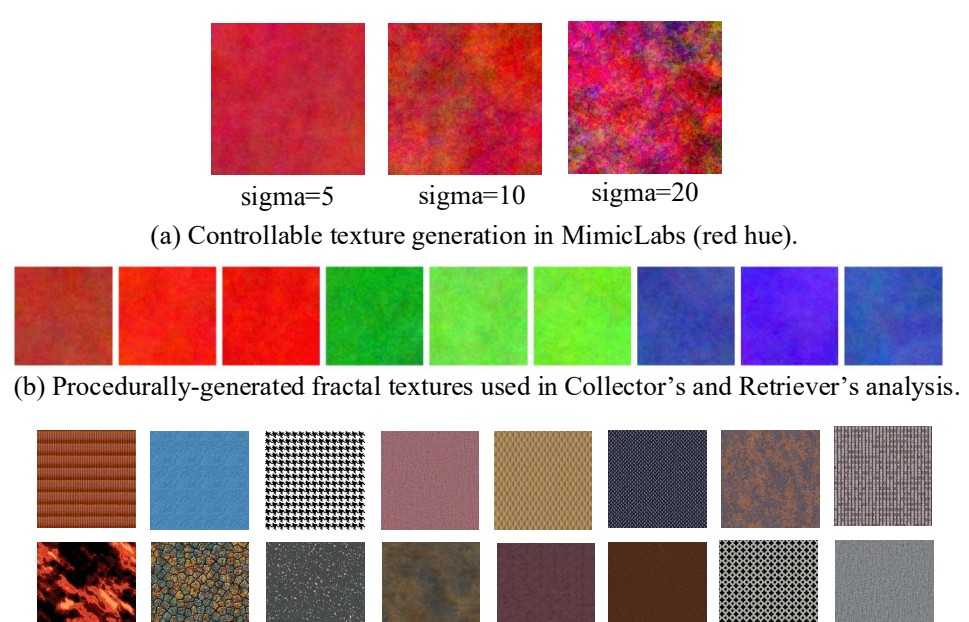

sigma=5  sigma=10  sigma=20

(a) Controllable texture generation in MimicLabs (red hue).

(b) Procedurally-generated fractal textures used in Collector's and Retriever's analysis.

(c) Sample irregular textures available to use in MimicLabs.

Figure 7: Procedural texture generation in MimicLabs. Fractal textures were used in the Collector's and Retriever's analysis when controllably generating target and co-training datasets with varying object textures. We also provide various other irregular textures that can be used for fabric, cookware, appliances, etc. for creating diverse scenes and tasks within MimicLabs.

# E   TRAINING AND EVALUATION DETAILS

**Simulation Policy Training.** While there are various approaches to imitation learning in robotics, one of them being waypoint prediction in the robot's base frame and relying on an off-the-shelf motion planner to fill in actions (Zeng et al., 2020; Shridhar et al., 2022; Saxena et al., 2024), we choose an end-to-end approach to policy execution predicting sequences of actions (at 20Hz) and executing them in a closed-loop fashion. We use BC-RNN (Mandlekar et al., 2021) and Diffusion Policy (Chi et al., 2023) to train imitation learning policies in our experiments, and use ResNet18 as the visual backbone. We also condition our policies on language instructions using FiLM layers in the vision backbone. We train all models for simulation for 500 epochs (250k gradient steps) using the Adam optimizer (Kingma & Ba, 2014) using learning rate $1e-4$. All our co-training experiments used 10 target demonstrations and 1000 co-training demonstrations (unless otherwise specified). We evaluate checkpoints every 50 epochs and report the peak success rate. Each evaluation is obtained by rolling out the policy 30 times.

Table 4: BC-RNN Policy Hyperparameters for Simulation Experiments

| | |
|---|---|
| batch size | 32 |
| sequence length | 10 |
| optimizer | Adam |
| learning rate | 1e-4 |
| image encoder | ResNet18 |
| image resolution | (128, 128) |

Table 5: BC-Transformer Policy Hyperparameters for Simulation Experiments

| | |
|---|---|
| batch size | 32 |
| context length | 10 |
| embed dim | 512 |
| num layers | 6 |
| num heads | 8 |
| embedding dropout | 0.1 |
| attention dropout | 0.1 |
| optimizer | Adam |
| learning rate | 1e-4 |
| image encoder | ResNet18 |
| image resolution | (128, 128) |

**Real-World Policy Training and Evaluation Details.** We use Diffusion Policy (Chi et al., 2023) for all our real-world experiments. Our low-dimensional observations for all models were the end-effector position and orientation (represented as a quaternion). For our collector experiments, we used a single camera (out of the four external options) for the image observations and for the retriever experiments, we used both of the shoulderview cameras. We use random cropping on our image observations before passing them into a ResNet-18 visual encoder and apply a Spatial Softmax (Levine et al., 2016) on the encoder outputs. These features are concatenated with the low-dim observations, processed by an additional observation processing MLP and passed into a U-Net diffusion head which predicts actions. The hyperparameters for our model are listed in Table 6.

For evaluation, we trained our models for 600 epochs for the collector experiments and for 300 epochs for the retriever experiments. Empirically, we found no difference in performance between models trained for 300 epochs and those trained for 600. We perform 20 rollouts with each model and report the success rates.

## E.1   DATASET RE-BALANCING

Throughout our study, we assume $N_C \gg N_T$, i.e. the number of co-training demonstrations is much larger than that collected by the practitioner for the target environment and task. Training a BC policy using a naive combination of these two datasets could cause the policy to ignore the

Table 6: Diffusion Policy Hyperparameters for Real Robot Experiments

| | |
|---|---|
| batch size | 128 |
| observation horizon (To) | 2 |
| action horizon (Ta) | 8 |
| prediction horizon (Tp) | 16 |
| diffusion method | DDIM |
| optimizer | Adam |
| learning rate | 1e-4 |
| image encoder | ResNet18 |
| image resolution | (128, 128) |

target dataset altogether while still minimizing its training objective (Hejna et al., 2024), resulting in unstable training and bad downstream performance (Du et al., 2023). Therefore, we instead create a dataset that samples from the distribution $\mathcal{D}_{T,C}(\omega) \triangleq \omega \mathcal{D}_T + (1-\omega)\mathcal{D}_C$ where $\omega \in [0,1]$. We implement this using a sampler that creates training batches by retrieving demonstration trajectories with probability $\omega$ from $\hat{\mathcal{D}}_T$ and $(1-\omega)$ from $\hat{\mathcal{D}}_C$. We represent this weighted combined dataset as $\hat{\mathcal{D}}_{T,C}(\omega)$. For all our experiments, we keep $\omega = 0.5$ and try ablations with $\omega = 0.3$ and $\omega = 0.7$ on the retriever experiment (see Fig. 13).

# F    TASK DESCRIPTIONS

## F.1    MIMICLABS TASKS

The MimicLabs dataset contains over 3000 task instances encompassing different motion primitives for solving different tasks, varying camera poses, objects, table textures, spatial arrangements, and background scenes. We summarize the tasks included in the benchmark below:

1. `pick X` and `place` it in the bin (7 instances per lab)
2. `open Y` (2 instances per lab)
3. `close Y` (2 instances per lab)
4. `open X`, `pick Y` and `place` it in `X` (14 instances per lab)
5. `pick X`, `place` it in `Y` and `close Y` (14 instances per lab)
6. `turn on` stove
7. `turn off` stove
8. make coffee

where `X` can be replaced by one of 7 distinct objects available in each lab for data collection and policy evaluation, `Y` can be a drawer or a microwave, distinct instances of which are available in each lab. In total, there are ∼290 unique task instances in each lab, with skill-level overlap designed to test positive retrieval strategies. Additionally, for each task instance, we created multiple variations in camera poses (5), object and spatial arrangements (∼90 combinations), which create  450 task instances in each lab, totaling to over 3000 instances across 8 labs.

The following tasks from MimicLabs were used for experiments in Section 5.3, in increasing order of hardness:

- *bin carrot*: easy binning task with a non-precise object.
- *bin bowl*: binning task with harder and multi-modal grasping strategy in expert demos.
- *clear table*: the robot should `pull` open the top drawer of the cabinet and `pick-place` the bowl in it.
- *microwave teapot*: the robot should `pick-place` a teapot into the microwave and `close` the microwave door.
- *make coffee*: the robot should `pick` up the coffee pod and `place` it in the coffee machine and `close` its lid; only one lab has a coffee machine and so we cannot retrieve any skills pertaining to placing the pod in the coffee machine or closing its lid.

Table 7: Task instances evaluated in the MimicLabs benchmark results with corresponding camera pose and spatial arrangement configs.

| Task | camPose | objSpat | recepSpat |
|---|---|---|---|
| **bin carrot** | shoulder-left | left 20cm×20cm | right |
| **bin bowl** | shoulder-right | right 20cm×20cm | left |
| **clear table** | agent-front | left 20cm×20cm | right |
| **microwave teapot** | shoulder-right | right 20cm×20cm | left |
| **make coffee** | shoulder-right | center 20cm×20cm | left |

## F.2    TASK DETAILS FOR COLLECTOR'S EXPERIMENT IN SIMULATION

When analyzing dataset composition from a collector's perspective, we construct multiple variations of the *clear table* task. The *baseline variation* in the task consisted of a fixed agent-front camera pose, single hue (red) object textures, single hue (wooden/beige) table texture, and small (10cm×10cm) spatial arrangement for the object around the center of the table.

## F.3 REAL ROBOT TASKS

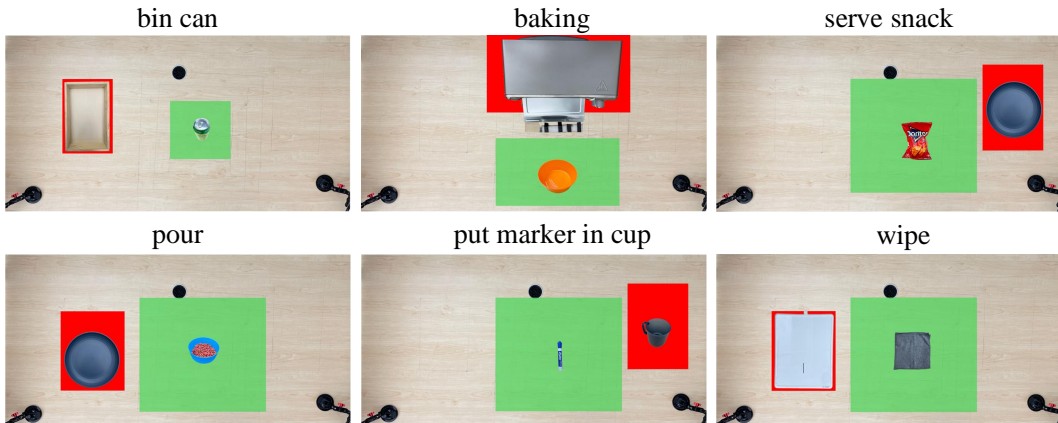

Figure 8: The green and red bounding boxes define the reset distributions for the object and the receptacle in each retriever task.

For our real-world experiments, we evaluate a variety of challenging and diverse table-top manipulation tasks that comprise multiple motion primitives and task horizons. Our goal was to ensure that our takeaways were broadly applicable to different end-to-end robot learning tasks. The reset distributions for the different objects in our retriever experiments are shown in Figure 8. During data collection, we uniformly distributed the object and the receptacle in their respective reset distributions. In evaluation, we uniformly sampled 20 poses for the objects and fixed these poses for testing all variations of each task. Below, we provide the descriptions and success criteria for each real-world task:

***store screwdriver***: this medium-horizon task consists of picking up a screwdriver, placing it in a drawer and closing the drawer. The location of the screwdriver varies but the drawer is fixed. **Success** is defined as the screwdriver in the drawer and the drawer closed.

***bin can***: this task involves one motion primitive where the robot must pick up a can from the table and place it in a bin. The location of the can varies, and the location of the bin is fixed. **Success** is defined as the can sitting upright in the bin.

***baking***: this is a challenging long-horizon task where the robot has to pick up a bowl and place it inside a toaster oven, push the oven tray, and close the oven. The location of the bowl varies but the oven is fixed. **Success** is defined as the bowl on the tray and the oven completely closed.

***serve snack***: the robot must pick up a red snack packet and put it on a plate. The locations of both the snack and the plate vary. **Success** is defined as the snack completely on the plate.

***pour***: the robot has to pick up a bowl containing beans, pour them onto a plate, and return the bowl to the workspace. **Success** is defined as all beans in the plate and the empty bowl being returned to the workspace.

***put marker in cup***: for this task, the robot should pick up a marker and place it in a cup. The locations of both the marker and the cup vary. **Success** is defined as the marker in the cup.

***wipe board***: the robot should pick up a cloth towel and use it to wipe off a 5cm mark on a whiteboard. The location of the towel varies, but the whiteboard and the mark are fixed. **Success** is defined as having at least half of the mark erased.

# G  NUMBER OF CO-TRAINING DEMOS FOR RETRIEVAL EXPERIMENTS ON MIMICLABS

Table 8: Number of demonstrations retrieved for each retrieval experiment on the MimicLabs dataset.

| Target task | Retrieving relevant object/skill | | | | | Missing relevant object/skill | | | | |
|---|---|---|---|---|---|---|---|---|---|---|
| | *obj/skill* | *+ camPose* | *+ objSpat* | *+ recepSpat* | *+ all* | *no-obj/skill* | *+ camPose* | *+ objSpat* | *+ recepSpat* | *+ all* |
| **bin carrot** | 24000 | 4800 | 6000 | 12000 | 1200 | 153800 | 30400 | 38600 | 77200 | 7600 |
| **bin bowl** | 28000 | 5600 | 7000 | 14000 | 1400 | 154200 | 30400 | 38800 | 77000 | 7600 |
| **clear table** | 46000 | 9200 | 22000 | 23000 | 3000 | 161800 | 32400 | 40600 | 81200 | 8000 |
| **microwave teapot** | 42200 | 8000 | 7400 | 14800 | 800 | 166600 | 32800 | 41400 | 82800 | 8200 |
| **make coffee** | 28000 | 5600 | 14000 | 14000 | 1400 | 158200 | 31400 | 79200 | 79000 | 7800 |

## H  REAL EXPERIMENT SETUP

We design our hardware setup to match that of DROID (Khazatsky et al., 2024) – a Franka robotic arm with a Robotiq gripper, mounted on a mobile platform, with several externally mounted cameras. Notably, our specific robot setup did **not** participate in the DROID dataset collection, minimizing the chance of data pollution. We use a Franka Research 3 7-DoF robotic arm with a Robotiq 2F-85 gripper. The robot is mounted on a mobile, height-adjustable platform. As illustrated in Figure 9, attached to the platform are two Zed 2 stereo cameras, and we also attach a Zed Mini camera to the wrist of the robot. Additionally, for our collector experiments, we attached two RealSense D435 cameras to the robot workspace for a total of four external cameras and one eye-in-hand camera. During data collection, all five camera streams are recorded and synchronized to the robot's actions. We additionally record robot proprioceptive information which includes joint positions, gripper positions, and end effector poses. The action space of the robot consists of an end-effector position (3-dimensional), rotation encoded in axis angles (3-dimensional), and gripper action (open/close). For teleoperating the robot and providing demonstrations, we use a Meta Quest 2 headset and controller.

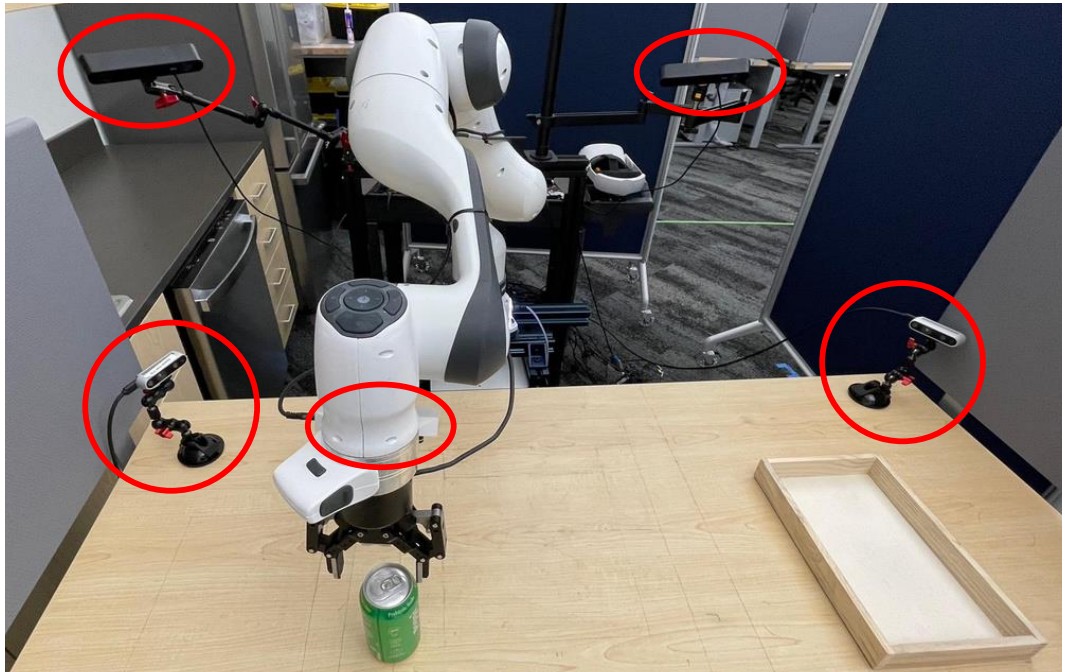

Figure 9: **Hardware setup.** The red circles show the camera set-up in the real experiment.

# I  REAL COLLECTOR DATASETS

**# Co-training and target demos:** for the *bin can* and *baking* tasks, we used 20 target demos for every model while for *store screwdriver* we used 10 target demos. All models were co-trained with 100 demos.

**Baseline dataset:** the baseline co-training dataset had minimal variation in each DV, namely, it consisted of a single camera view, a single texture of the target object, and a small reset distribution (25x25cm).

**Co-training dataset construction:** the co-training datasets were created by sampling evenly from the ranges of a particular DV. For example, in a *camPose* experiment where the co-training dataset was comprised of two cameras that are not target camera, the co-training dataset would be constructed by selecting 50 demos from one camera and 50 from the other, for a total of 100.

The per-DV variation for our collector experiments is:

1. *camPose* : we select one of the 4 external cameras as the target camera during evaluation and co-train models using various combinations of the cameras.

2. *objTex* : we pick one color of the target object for evaluation, and co-train models with demos of varying target object textures. Note that we do not ablate on material properties of these objects. We also keep our lighting conditions consistent within each dataset to remove any variations in object visuals due to lighting-material interactions.

3. *objSpat* : we pick a spatial distribution for the target object during evaluation and co-train models with demos where the target object has varying spatial distributions. The different co-training spatial distributions are concentric boxes that increase in size until they cover the target spatial distribution completely.

**DV data collection:** to build the datasets for each DV, we had to vary the setup during demonstration collection. Specifically, for:

1. *camPose* : we collected 100 demos of the task, with a small reset distribution and a single texture. Since all four external cameras were streaming simultaneously, we could build co-training datasets with combinations of the different cameras through post-processing.

2. *objTex* : we used the data collected in 1. for our baseline dataset. We then collected an additional 50 demos with a second color to build our second co-training dataset (50 from each) and so on with a third color. All demos were collected with a small spatial distribution.

3. *objSpat* : again, we used the data collected in 1. for our baseline dataset. We then collected enough demos to cover a medium reset distribution of 38x38cm and even more demos to cover a large reset distribution of 50x50cm. Each co-training dataset (small, medium, large) consisted of demos sampled evenly from each distribution. The color of the target object remained fixed and we trained and evaluated on a single camera pose.

## J    DROID RETRIEVAL AND METADATA PROCESSING

In order to perform retrieval on DROID, we had to create additional metadata that would allow querying along specific DVs. Here we explain the creation of this metadata:

- **Target Object**: DROID has up to three different language instructions describing the task executed in each demonstration. To identify the target object from the language instructions, we first merged the multiple instructions into a single coherent command for each demo. After combining the instructions, we applied syntactic parsing, focusing specifically on the direct and indirect objects of the action verbs in the combined instruction. Once the direct and indirect objects were extracted, we performed agglomerative clustering, using cosine similarity between the word embeddings of these objects. The object with the highest similarity within the primary cluster was designated as the target object. If multiple objects were mentioned, priority was given to the first occurrence in the instruction.

- **Object Spatial**: to get the location of the manipulated object, we used the gripper state of the robot as a heuristic. We determined when the gripper closed and used the position of the end-effector at this timestep as the position of the manipulated object. If the gripper closed and opened multiple times during the demo, we used the first gripper close for the position of the object. The gripper state was averaged over a window of 15 timesteps to filter out accidental gripper closes/opens.

- **Object Colors**: to classify the color of the target object in the demos, we first saved the initial image frame from each demo. We then used the LLaVA v1.5 7b model (Liu et al., 2024) to detect the color of the target object from this image. The prompt provided to the model was: "USER: What's the color of the <target object> in the image? Answer in just one adjective word. ASSISTANT:".

The different types of retrieval we performed on DROID are: *obj/skill*, *obj/skill + camPose* , *obj/skill + objTex* , and *obj/skill + objSpat* . The process of retrieving these DVs is as follows:

- *obj/skill*: using the language instructions in DROID, we filtered out demos where our target object was being manipulated.

- *obj/skill + camPose* : from demos containing the target object, we further filter the demos where the camera positions are within 20cm of the target camera in the X- and Y-axes, and 10cm in the Z-axis. This is done using the per-demo camera extrinsic metadata provided in DROID.

- *obj/skill + objTex* : after filtering out demos with the target object, we further filter by selecting only demos where the color is the same as our testing setup.

- *obj/skill + objSpat* : from demos containing the target object, we select the demos where the spatial distribution of the object is a 60x60x30cm cuboid centered at the middle of our testing distribution. The maximum size of our testing distribution was 50x50x1cm so the retrieved spatial distributions would always cover the testing distributions.

Examples of demos after retrieval are shown in Figures 10, 11, and 12.

Target                                    Retrieved

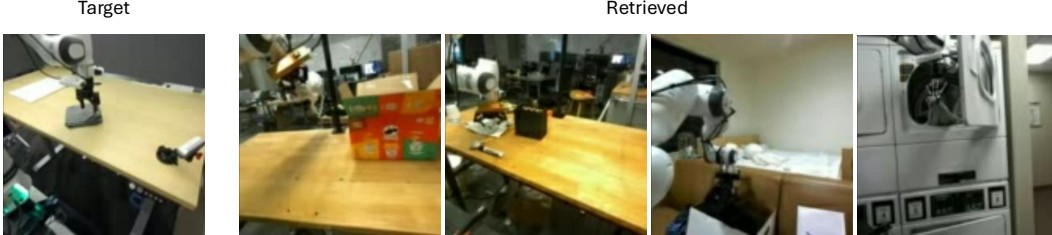

Figure 10: **Retrieved cotraining demos for the *wipe board* task with the camera pose aligned.** We can see that the spatial locations and textures of the objects in the demos are not necessarily aligned with our target object. However, the camera pose is the same.

Target                                    Retrieved

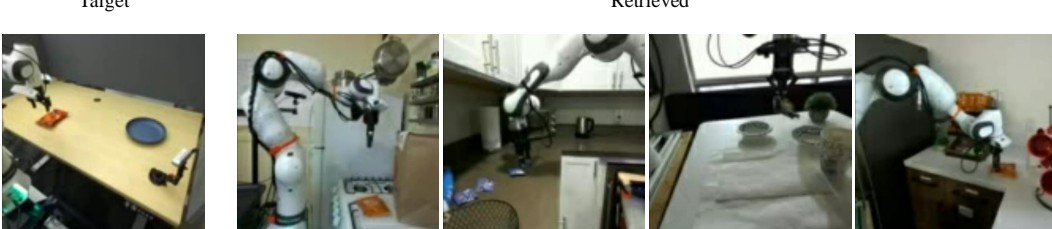

Figure 11: **Retrieved cotraining demos for the *serve snack* task with the object spatial location aligned.** The camera poses and textures seen in the demos are not the same as our target task, but the location of the object (in front of the robot and level with its base) is similar.

Target                                    Retrieved

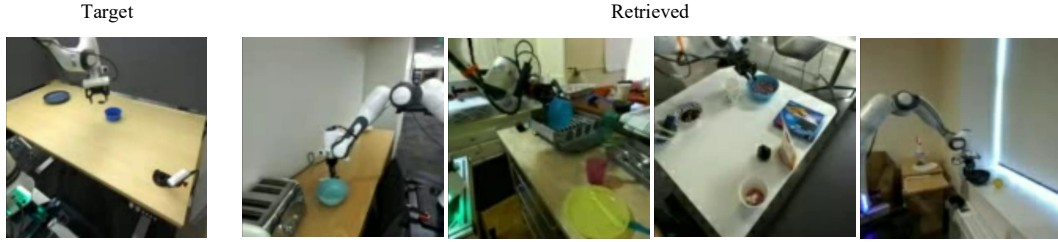

Figure 12: **Retrieved cotraining demos for the *pour* task with the object texture aligned.** The camera poses and spatial locations of the target object are not the same, but the color is aligned.

## K    ADDITIONAL RESULTS

Table 9: **Analyzing the effects of misaligned and diversities in DVs from a collector's perspective.** Success rates on the *make coffee* task with different variations in co-training distributions. Target variations are perturbations to one DV while others stay fixed and perfectly-aligned with the baseline distribution. Success rates highlighted when boost in performance is $\geq 10\%$ compared to baseline.

| **DV** w/ target-cotrain misalignment | **Target only** | **Co-training with different DV distributions** | | | | |
|---|---|---|---|---|---|---|
| | | Baseline | camPose | objTex | tableTex | objSpat |
| camPose | 10 | 16.67 | **43.33** | 3.33 | 3.33 | 16.67 |
| objTex | 10 | 6.67 | 6.67 | **36.67** | **36.67** | **43.33** |
| tableTex | 10 | 16.67 | 10 | 26.67 | **30** | **33.33** |
| objSpat | 16.67 | 10 | 10 | 10 | 20 | **26.67** |

Table 10: **Motion diversity in co-training.** We tested two variations of the *clear table* task: easy (*put the bowl in the drawer and close it*) and hard (*open the drawer and put the bowl in it*). The easy variant requires learning the `push` and `pickPlaceTopDrawer` motion primitives, while the hard variant requires `pull` and `pickPlaceTopDrawer`. Co-training datasets (left to right) are ordered in increasing order of motion diversity they bring for co-training (500 demos per primitive), with the most diverse dataset covering all the motion primitives needed to solve the target task but with increased heterogeneity in robot motion.

| **Target variation** | **Target only** | **Co-training with different motion primitives** | | | | |
|---|---|---|---|---|---|---|
| | | pull | push | pull+ pickPlaceBasket | push+ pickPlaceBasket | pull+push+pickPlaceBasket+ pickPlaceTopDrawer |
| `pickPlaceTopDrawer+ push` | 63.33 | 43.33 | 56.67 | 63.33 | 63.33 | **83.33** |
| | | pull | push | pull+ pickPlaceBasket | push+ pickPlaceBasket | pull+push+pickPlaceBasket+ pickPlaceTopDrawer |
| `pull+ pickPlaceTopDrawer` | 23.33 | 43.33 | 16.67 | 36.67 | 43.33 | **50** |

Table 11: **MimicLabs Retrieval.** Task success in the MimicLabs benchmark when co-training using various retrieval strategies, using BC-Transformer policy. Success rates averaged over 30 rollouts, and max across 500 epochs of training. Results in **bold** are best co-training within the sub-experiment. We provide the number of demonstrations retrieved for each experiment in Table 8.

| Target task | #demos | Target only | Retrieving relevant object/skill | | | | | Missing relevant object/skill | | | | |
|---|---|---|---|---|---|---|---|---|---|---|---|---|
| | | | ✓ *obj/skill* | *+ camPose* | *+ objSpat* | *+ recepSpat* | *+ all* | ✗ *obj/skill* | *+ camPose* | *+ objSpat* | *+ recepSpat* | *+ all* |
| **bin carrot** | 10 | 53.33 | 90 | **93.33** | 83.33 | **93.33** | **93.33** | 50 | 43.33 | 53.33 | **73.33** | 40 |
| **bin bowl** | 10 | 36.67 | 56.67 | 50 | 50 | **63.33** | 56.67 | 53.33 | **53.33** | 36.67 | 43.33 | 30 |
| **clear table** | 20 | 30 | 40 | **53.33** | 33.33 | 23.33 | 43.33 | 13.33 | 23.33 | 23.33 | 23.33 | **33.33** |
| **microwave teapot** | 20 | 10 | 13.33 | 13.33 | 20 | 16.67 | **30** | 16.67 | **23.33** | 6.67 | 3.33 | 10 |
| **make coffee** | 50 | 6.67 | 10 | **20** | 10 | 16.67 | 13.33 | 10 | **20** | 3.33 | 3.33 | 13.33 |

Table 12: **MimicLabs retrieval in a pretrain-finetune setup.** Task success in the MimicLabs benchmark when pre-training using various retrieval strategies and fine-tuning on the target dataset. Success rates averaged over 30 rollouts, and max across 500 epochs of training. Results in **bold** are best co-training within the sub-experiment. We provide the number of demonstrations retrieved for each experiment in Table 8.

| Target task | #demos | Target only | Retrieving relevant object/skill | | | | | Missing relevant object/skill | | | | |
|---|---|---|---|---|---|---|---|---|---|---|---|---|
| | | | ✓ *obj/skill* | *+ camPose* | *+ objSpat* | *+ recepSpat* | *+ all* | ✗ *obj/skill* | *+ camPose* | *+ objSpat* | *+ recepSpat* | *+ all* |
| **bin carrot** | 10 | 50 | 76.67 | 80 | 83.33 | **86.67** | 83.33 | 60 | 73.33 | **80** | 63.33 | 76.67 |
| **bin bowl** | 10 | 33.33 | 76.67 | 73.33 | 66.67 | 66.67 | **83.33** | 60 | 66.67 | 66.67 | **73.33** | 60 |
| **clear table** | 20 | 36.67 | 23.33 | **60** | 20 | 20 | 46.67 | 43.33 | 30 | 26.67 | 46.67 | **50** |

Table 13: **Evaluating the importance of alignment and diversity when co-training for a target task in the collector setup (real-world).** The grey circles above each column represent the entire possible distribution for that DV. The **green** circle is the **target** distribution and the **red** circles are the distributions found in the **cotraining** datasets. For the *camPose* setup, this involves co-training with 1 or 2 cameras that are not the target camera and then co-training with all the cameras as well as only the target camera. For *objTex* experiments, we cotrain with 1/2/3 colors that are all different from the target color. For the *objSpat* experiments, we increase the reset distribution of the object in the co-training datasets (*small < medium < large*) until the co-training dataset covers all of the target reset distribution.

| Task | camPose | | | | | objTex | | | | objSpat | | | |
|---|---|---|---|---|---|---|---|---|---|---|---|---|---|
| *bin can* | 50 | 60 | 40 | 55 | **75** | 50 | 90 | **95** | 90 | 10 | 40 | 50 | **70** |
| *store screwdriver* | 10 | 55 | 50 | 50 | **60** | 5 | 25 | 20 | **70** | 5 | 15 | **45** | 30 |
| *baking* | 40 | 70 | 70 | 80 | **85** | 60 | 95 | **100** | 95 | 45 | 75 | 80 | **85** |

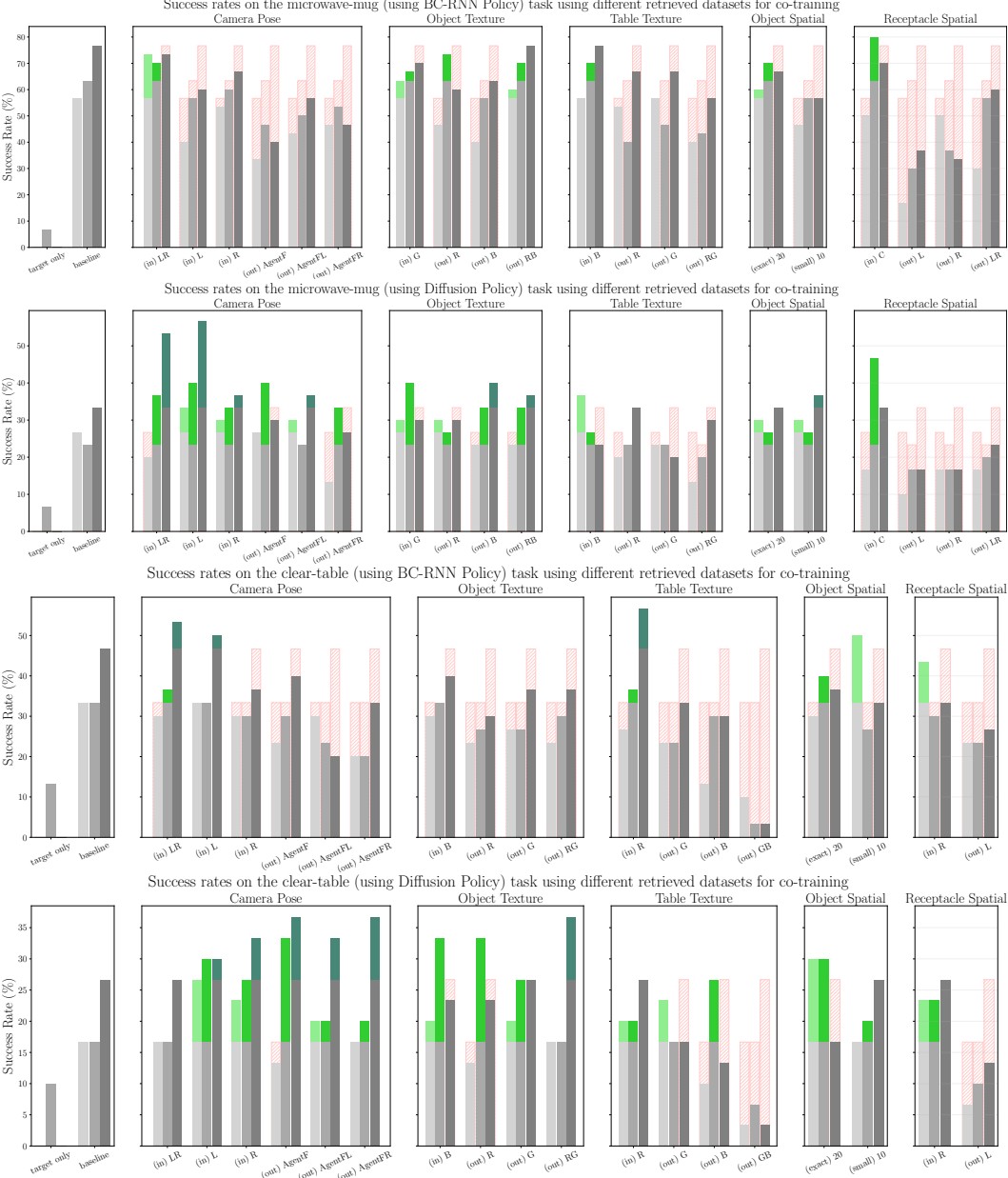

Figure 13: Results showing success rates on two tasks using BC-RNN and Diffusion Policy for different retrieved datasets along all considered DVs. The DVs (left to right) are camera pose, object texture, table texture, object spatial, and receptacle spatial arrangements. The three bars (left to right) for each experiment use $\omega = 0.7$, $\omega = 0.5$, and $\omega = 0.3$ respectively (see Appendix E.1 for dataset re-balancing details). Portions of bars in shades of green shows performance improvement over baseline co-training, while hollow red portions depict performance degradation.

