# OpenReview forum: "What Matters in Learning from Large-Scale Datasets for Robot Manipulation"
_ICLR.cc/2025/Conference — ICLR 2025 Poster_

### Official Review · Reviewer_CfcJ · 2024-10-20

**Soundness:** 3
**Presentation:** 2
**Contribution:** 3
**Rating:** 6
**Confidence:** 5

**Summary:**

This paper explores the key factors that influence the success of robot manipulation policies trained on large-scale datasets. It introduces a synthetic data generator, MimicLabs, to emulate common variations in real-world datasets such as camera poses and object arrangements. The study focuses on identifying the types of diversity that should be emphasized when collecting new datasets or retrieving relevant demonstrations from existing ones to maximize downstream task performance. Using simulation and real-world tasks, the authors demonstrate that factors like camera pose and spatial arrangement are critical to both dataset diversity and alignment. These insights are validated through experiments on the DROID dataset, leading to improvements in policy learning by up to 70%.

**Strengths:**

1. The paper tackles an important problem—optimizing large-scale datasets for robotic manipulation—by systematically analyzing how various dimensions of variation (DVs) in the data affect learning. It provides a novel approach through its MimicLabs synthetic data generator, which offers a controlled environment for testing different dataset compositions, addressing a key gap in robot learning research.
2. This work has the potential to significantly impact the future of large-scale robot dataset collection and utilization. By offering actionable insights on how to prioritize data collection and improve policy learning through targeted dataset retrieval, it lays the foundation for more efficient robot learning pipelines, reducing the cost and complexity of data collection efforts.

**Weaknesses:**

1. Some conclusions are well-known: The paper emphasizes that “camera poses and spatial arrangements are crucial dimensions for both diversity in collection and alignment in retrieval,” which is somewhat redundant with existing research. Camera pose, in particular, has long been recognized as a challenging factor to generalize. The generalization issues of spatial arrangements are also quite intuitive since they are linked to training distribution. If the training distribution is narrow, out-of-distribution (OOD) failure at test time is expected.
2. Dataset Dependency: The experiments and conclusions are closely tied to the DROID dataset, raising concerns about their generalizability to other robotic platforms or datasets. More varied datasets should be included in future experiments to ensure the robustness of the insights across different domains.
3. Difficulty in real-world implementation of “Dimensions of Variation” (DVs): While the paper proposes the concept of DVs, it might be challenging to apply this approach in real-world scenarios where it is impractical to enumerate all possible variations. For example, factors such as object material or lighting conditions are not sufficiently addressed. Furthermore, real-world robot performance is often contingent on handling the complex interplay of multiple variations simultaneously rather than generalizing to each factor individually.

**Questions:**

1. Have you tried using only wrist cameras? One question that could have been explored more thoroughly is whether using dynamic wrist cameras, which vary poses constantly, would act as a strong form of data augmentation and mitigate the problem of overfitting to fixed camera poses. If this technique were used, it could potentially reduce sensitivity to specific camera poses and improve generalization.
2. Generalization beyond single variation dimensions: The current analysis focuses on individual DVs (like camera pose or object arrangement), but in the real world, robots often face scenarios where multiple variations occur simultaneously. Have you considered experiments that explore the interaction effects between multiple DVs, and if so, how do the insights change?

---

> ### Author Response · Authors · 2024-11-23
> **Response to Reviewer Cfcj (1/3)**
>
> We thank the reviewer for their thoughtful review of our work, and recognizing the potential of this work to significantly impact future data collection and utilization. Please find our response to their concerns below.
>
> > Some conclusions are well-known: The paper emphasizes that “camera poses and spatial arrangements are crucial dimensions for both diversity in collection and alignment in retrieval,” which is somewhat redundant with existing research. Camera pose, in particular, has long been recognized as a challenging factor to generalize. The generalization issues of spatial arrangements are also quite intuitive since they are linked to training distribution. If the training distribution is narrow, out-of-distribution (OOD) failure at test time is expected.
>
> Thank you for raising this point. Although much of the data collection effort in robot learning has been guided by anecdotal knowledge regarding the significance of camera poses and spatial arrangements among other DVs [1,2,3], there remains a lack of a systematic framework to conduct a thorough and nuanced analysis of how to optimize data collection efforts or effectively leverage large-scale datasets. Our paper introduces a *structured methodology,* delineating four distinct scenarios of **diversity / alignment** between target and co-training datasets (as illustrated in **Figure 2**), which encompass two analytical perspectives: that of the data collector and a retriever. Our findings are systematic, reproducible, and extendable beyond the scope of this study using the conceptual methodology and software framework we provide. We systematically show using a reproducible experimental framework which DVs are meaningful for adding diversity during data collection, as well as how to best retrieve datasets from large demonstration pools. Our framework is *general-purpose,* and can be utilized by future work to analyze even more DVs and their distributions that can enable efficient allocation of data collection effort in robotics.
>
> Furthermore, one of the reasons we speculate it is challenging to learn meaningful skills from DROID [3] for new downstream tasks (as seen in our own co-training results) is because it is hard to find relevant demonstrations that align with the downstream camera pose, making a large section of the dataset not useful when the downstream camera pose is very different from most demonstrations in the dataset. If our takeaways were given beforehand, data collection in DROID could have been more sparse or dense along different DVs, allocating data collection budget where it was more necessary.
>
>
> > Dataset Dependency: The experiments and conclusions are closely tied to the DROID dataset, raising concerns about their generalizability to other robotic platforms or datasets. More varied datasets should be included in future experiments to ensure the robustness of the insights across different domains.
>
> While we agree that a singular dataset may contain bias, DROID is the culmination of over a year of in-the-wild data collection from more than 50 individuals, in 52 different buildings, and covering 86 different tasks. DROID also has very specific data collection protocols to reduce bias (for example, the GUI randomly samples tasks and prompts the user for scene augmentations). The robotics community is moving towards collecting larger datasets, but at the moment no other imitation learning dataset matches DROID’s diversity, scale, and reproducibility of the robot setup. The other dataset that may seem comparable to DROID is the Open X-Embodiment dataset [4], however this is a compilation of very diverse datasets, all using different platforms, robots, and demonstration collection protocols. While interesting, we leave the study of what matters in learning from large scale cross-embodiment datasets to future work. Considering we are evaluating on a single platform (which is very common in the robotics community) and our needs for scale and diversity, we believe that the DROID dataset is a valid choice for our experiments. Furthermore, as future large scale datasets become available, our framework for analysis will continue to be useful for the robot learning community.
>
> We believe our conceptual analysis framework and the tools to generate synthetic data for systematic simulation-based study can be broadly applicable for new settings and datasets when available. Therefore we believe our work will be of wide interest to the robot learning community and beyond.

---

> ### Author Response · Authors · 2024-11-23
> **Response to Reviewer Cfcj (2/3)**
>
> > Difficulty in real-world implementation of “Dimensions of Variation” (DVs): While the paper proposes the concept of DVs, it might be challenging to apply this approach in real-world scenarios where it is impractical to enumerate all possible variations. For example, factors such as object material or lighting conditions are not sufficiently addressed.
>
> Thank you for your insightful comment. We agree that it is difficult to enumerate all possible DVs in real-world scenarios. However, we have shown that the DVs we analyze in this paper are *measurable* both in simulation and in meta-data from real-world demonstrations (e.g., object configuration, camera poses, obtained from the DROID dataset). Lighting conditions, for example, are hard to measure in the real world, and hence it was not meaningful to generate simulation data controlling this factor.
>
> Furthermore, our analysis framework helps identify learning-relevant DVs, which can serve as a codebook for future data collectors. We acknowledge that it takes effort to measure, vary, and then record appropriate meta-data for a DV during robot data collection in the real world. Our paper helps demonstration collectors understand where this effort is best spent. In a large-scale data collection effort such as DROID, the concept of DVs is already implicitly implemented. Collectors are randomly asked to switch tasks, move their camera, change the items, or adjust the lighting. But it is unclear what types of changes to make, how frequently, and to what degree. Our paper helps shed light on these practical considerations as future data collection protocols necessarily deal with the difficulties of DV enumeration.
>
>
> > Have you tried using only wrist cameras? One question that could have been explored more thoroughly is whether using dynamic wrist cameras, which vary poses constantly, would act as a strong form of data augmentation and mitigate the problem of overfitting to fixed camera poses. If this technique were used, it could potentially reduce sensitivity to specific camera poses and improve generalization.
>
> While we agree with the reviewer that using only wrist cameras could mitigate the camera viewpoint gap, our goal is *not* to find the best hardware/sensor configuration to achieve the best performance on specific tasks. Rather, the goal of this paper is to illustrate how the distribution of common sources of dataset variation (including static camera viewpoints) affects downstream policy performance. With this goal in mind, we make a good faith effort to utilize state of the art techniques and policies in our study. As pointed out in your review, there are certainly many techniques for adding different types of inductive biases to the learning process, such as training using only wrist cameras. For the sake of time and resources, we leave these experiments to future research and our paper provides the tools and the framework for such studies. We stick to a default setting where we use both agentview and wrist cameras for all our experiments in simulation. Furthermore, training on multiple camera views is common practice in the robot learning community with external/global camera views being beneficial to guide the robot through the full task plan as many things may be out of view of the wrist camera.

---

> ### Author Response · Authors · 2024-11-23
> **Response to Reviewer Cfcj (3/3)**
>
> > Generalization beyond single variation dimensions: The current analysis focuses on individual DVs (like camera pose or object arrangement), but in the real world, robots often face scenarios where multiple variations occur simultaneously. Have you considered experiments that explore the interaction effects between multiple DVs, and if so, how do the insights change?
>
> Thank you for your insightful question, which was also raised by Reviewer *Gs6e*. We agree that it would be interesting to see experiments that directly test how conflicting DVs affect target task success. For example, should we choose to diversify camera poses or spatial arrangements during data collection if only one of the two is possible?
>
> Our study provides some interesting conclusions even while analyzing one DV at a time. In the collector’s experiments, if we look at **Table 1** and **compare the relative increase in success rate** over the baseline dataset, we can infer what DV to prioritize in the case of conflict or when there is a fixed budget for data collection. Our results suggest that Cam Pose > Object Spatial > Table Texture > Object Texture. As validated in our results, we find these takeaways to hold in a retrieval setting, and even translated into real-world performance using retrieval from a large-scale real-robot dataset.
>
> Analyzing all possible combinations of DVs would vastly increase the number of experiments that need to be run. For example, determining whether to pick a demonstration that has aligned camera pose but misaligned spatial arrangements, or vice-versa, are just two of the N-factorial permutations of experiments that need to be run (where N is the number of DVs). Given that we analyze 5 DVs, this necessitates 120 different dataset compositions for one task and 1000s of GPU hours for inference. While we do not have the resources to run each of these experiments, our data generation framework and method of analysis make it very easy to design such experiments. We also look forward to seeing these results if future researchers have the computational resources to directly answer this question.
>
> References:
>
> [1] Bridge Data: Boosting Generalization of Robotic Skills with Cross-Domain Datasets, Ebert et al., 2021
>
> [2] RT-1: Robotics Transformer for Real-World Control at Scale, Brohan et al., 2023
>
> [3] DROID: A Large-Scale In-The-Wild Robot Manipulation Dataset, Khazatsky et al., 2024
>
> [4] Open X-Embodiment: Robotic Learning Datasets and RT-X Models, Open X-Embodiment Collaboration, 2023

---

> ### Author Response · Authors · 2024-12-02
>
> We thank the reviewer for their valuable time spent in reviewing our work, and raising thoughtful questions which we addressed in our rebuttal. As the discussion period comes to an end, we sincerely request the reviewer to let us know if there are any more pending questions from their end, and adjust their score to reflect their final rating of our work. Thank you!

---

### Official Review · Reviewer_D5QM · 2024-10-29

**Soundness:** 2
**Presentation:** 3
**Contribution:** 3
**Rating:** 6
**Confidence:** 3

**Summary:**

This work conducts a thorough investigation on creating and utilizing robotics dataset for co-training robot policy with imitation learning. It demonstrates the most useful types of diversity for dataset collection and proposes insightful yet effective retrieval methods from existing datasets to improve task performance.

**Strengths:**

1. This work tackles a popular yet significant problem concerning usage of large-scale robotics dataset.
2. The authors conduct extensive experiments on collector and retriever perspective with various DVs.
3. The paper is well-written and easy to follow.

**Weaknesses:**

1. The experiments in Section 5.1 only consider one task *clear table*, making the conclusions less convincing. Results on more simulation tasks or clarification on the representativeness of this task should be added.
2. The co-train setting in this paper largely focuses on using generated/retrieved data from the same task as the test task (except for some results in Table 3). However, aligning the task setting between the target task and the tasks in a public large-scale dataset is difficult in most cases, especially for real-world tasks.  More experiments and analysis on co-training with different tasks can add to the practicality of this work.
3. Some of the results may be confusing. Please refer to the Question section.

**Questions:**

1. In Table 1, it can be confusing why training and testing on only target distribution will yield such poor results, since overfitting to a rather small range of variation seems not to be that hard in imitation learning.
2. In Table 1, I am confused why co-training with varied camera poses can improve policy performance to such a great extent, because there are only several (~5) poses and they are very different. Maybe some videos and extra explanations can help clarification.
3. In Table 2 "Object spatial" part, why larger co-training distribution produces lower performance?
4. In Figure 5, I am curious whether the improved performance is attributed to the co-training setting or alignment between target and training task. For example, how is the performance if the policy is trained with retrieved Droid data without target data?

---

> ### Author Response · Authors · 2024-11-23
> **Response to Reviewer D5QM (1/3)**
>
> We thank the reviewer for their time spent in going through our paper, giving insightful comments, and acknowledging that we conduct extensive experiments and that our paper is well-written and easy to follow. Please find below our response to their concerns, including *additional results* in the revised paper.
>
> > The experiments in Section 5.1 only consider one task clear table, making the conclusions less convincing. Results on more simulation tasks or clarification on the representativeness of this task should be added.
>
> Thank you for raising this important point, which was also raised by Reviewer *Gs6e*. Our response is organized as follows:
> 1. We provide further details about the motivation behind the experiments from the collector's perspective, highlight the multi-faceted nature of the single domain considered in simulation (Section 5.1), and subsequent extensive evaluation on a real robot (**Table 13** in Appendix K).
> 2.  We run additional experiments on a second task (`make coffee`) and show the same conclusions hold (see **Table 9** in Appendix K for additional results).
>
> The experiments from the **collector's perspective** offer a way for data collectors to test the importance of each dimension of variation (DV) to inform where they should allocate data collection effort. This is a large multi-faceted experiment where we collect multiple datasets with exhaustive variations along 4 DVs of the target task (camera poses, object textures, table textures, and spatial arrangements of the object) as well as 4 different dataset variations for co-training. We believe this framework and our in-depth analysis of a meaningful long-horizon task is valuable for future data collectors. In addition to **Table 1**, we have more results from the collector’s perspective in Appendix K where we analyze the motion primitive DV for data collection in this task (**Table 10**).
> This experimental design was specifically chosen to be adaptable to real-world settings with minimal data collection effort. Hence we also conducted this experiment in the **real world** (summarized in Section 5.4.1 and **Table 13**) on **3 different tasks**, collected over **1200 teleoperated demonstrations** and evaluated **39 different models across 13 different target-cotraining dataset combinations.**
> To further support our claims, we have added another challenging task in simulation to this analysis, `make coffee` (put the coffee pod in the coffee machine and close the lid). We find this task to be meaningfully different from the `clear table` task, with significantly different geometries, requiring higher precision for pick and place, and even different articulation. We present the results below, and have added them to **Table 9** in Appendix K of the revised paper.
>
> | Target task variation &#8595; | target only	|baseline| camPose | objTex |tableTex |objSpat |
> |-|:-:|:-:|:-:|:-:|:-:|:-:|
> | camPose 		|10	|16.67	|**43.33**	|3.33		|3.33	|16.67	|
> | objTex 		|10	|6.67	|6.67		|**36.67**	|**36.67**|**43.33**|
> | tableTex 		|10	|16.67	|10	 	|26.67		|**30**	|**33.33**|
> | objSpat		|16.67	|10	|10	 	|10		|20	|**26.67**|
>
> Similar to the setting in Table 1, different columns represent different co-training distributions with high variation along one DV, stemming from the baseline variation. The target task variations (rows) also contain misalignment along one DV from the baseline as well as the corresponding high-variation co-training dataset. We note that many of our takeaways are still the same in this experiment, showing (1) misaligned camera poses between target and co-training can hurt skill transfer, (2) disparity in object textures between target and co-training can be mitigated by other DVs, and (3) both camera poses and spatial arrangements are crucial aspects in data collection.

---

> ### Author Response · Authors · 2024-11-23
> **Response to Reviewer D5QM (2/3)**
>
> > The co-train setting in this paper largely focuses on using generated/retrieved data from the same task as the test task (except for some results in Table 3). However, aligning the task setting between the target task and the tasks in a public large-scale dataset is difficult in most cases, especially for real-world tasks. More experiments and analysis on co-training with different tasks can add to the practicality of this work.
>
> Thank you for your comment. Both our simulated and real-world experiments do focus on co-training with different tasks, which as you pointed out, is an important practical aspect of such a study. Below we summarize our experimental methodology that starts with a constrained setup and then relaxes constraints to prove usefulness in real-world tasks.
>
> In this paper, we take a controlled approach to study the effects of different data compositions for co-training, along multiple DVs, and then validate our takeaways on large-scale simulated and real-world datasets that contain a **variety of tasks** that we retrieve from. We start by analyzing the effect of disparity between target and co-training datasets one DV at a time, while keeping the task same across both datasets. This allows us to draw conclusions about the importance of DV alignment and diversity, without any negative confounding effects due to task disparity. These experiments are shown in **Sections 5.1** and **5.2**. Next, we relax these constraints and validate our takeaways by studying retrieval on a large-scale simulated dataset (*MimicLabs*) in **Section 5.3**. Our simulated dataset contains a large variety of tasks to choose from across multiple scenes, each requiring a different `pick`, `place`, `open` or `close` action, or a combination of these for a longer horizon task, and including different object instances in each lab to diversify manipulation strategies. Finally, we mirror this study on a real robot using the large-scale DROID dataset in **Section 5.4.2**, where we show co-training on diverse tasks while retrieval on the proposed and analyzed DVs in previous sections. Overall, both our simulated and real-world experiments focus on co-training with different tasks, which as you pointed out, is an important practical aspect of such a study.
>
>
> > In Table 1, it can be confusing why training and testing on only target distribution will yield such poor results, since overfitting to a rather small range of variation seems not to be that hard in imitation learning.
>
> We agree with the reviewer that overfitting to a target dataset is an easy task for imitation learning, but high success is guaranteed only when the dataset is large enough, which is not a realistic setting. If we had access to a large set of target demonstrations ($\sim$500) for a task with the reset ranges we considered, a majority of tasks could be completed with >90% success rate, but this would not be the case when only 10 demonstrations are available. We are interested in studying the setting where people want to quickly train their robot with few (10-30) demonstrations on a new task to high success, by leveraging a large co-training dataset that has varying degrees of alignment with their task along different DVs, to boost downstream performance. Stemming from this practical motivation, we hope to answer the scientific question of how dataset composition affects downstream learning performance (in our case, co-training), which is of wide interest to the community.
>
> To further address your comment about low success with imitation learning using few target demonstrations, we want to point out that the reset distribution range for all tasks in our paper is higher than what is used in other related research works.  For example, in our pick and place tasks, the reset range is 0.2m x 0.2m. In LIBERO [1], which is a common benchmark in recent imitation learning papers [2,3], the reset range is 0.05m x 0.05m for most tasks. This is one of the major reasons why we do not see strong overfitting even with 50 demos for the tasks considered in our paper.

---

> ### Author Response · Authors · 2024-11-23
> **Response to Reviewer D5QM (3/3)**
>
> > In Table 1, I am confused why co-training with varied camera poses can improve policy performance to such a great extent, because there are only several (~5) poses and they are very different. Maybe some videos and extra explanations can help clarification.
>
> Thank you for raising this point for clarification. The results in Table 1 indeed show that adding variation in camera poses helps the robot achieve higher success when there is disparity along multiple DVs. We believe that training on diverse camera poses on these tasks allows the robot to see larger background variations, potentially leading to visual robustness and subsequently helping alleviate disparity between target and co-training datasets along textures. To further clarify the diversity of camera poses in training, we have **added details in Appendix C** about the exact distributions of camera poses used in MimicLabs, including **Figure 6** that illustrates these distributions around the table.
>
>
> > In Table 2 "Object spatial" part, why larger co-training distribution produces lower performance?
>
> Thank you for pointing this out. This is indeed a typo/error in Table 2’s column headers for the object-spatial DV that does not affect our conclusion - the columns in this table are in the order of (left to right) perfect alignment, followed by different counterfactual retrievals. For the object spatial DV, the order of graphical stamps was accidentally flipped. We have since updated Table 2 with this fix, and larger co-training distributions do indeed produce a higher performance.
>
>
> > In Figure 5, I am curious whether the improved performance is attributed to the co-training setting or alignment between target and training task. For example, how is the performance if the policy is trained with retrieved Droid data without target data?
>
> Thank you for your insightful question. We agree that such an experiment would be very meaningful to ascertain whether the performance improvements from different retrieval strategies can be attributed to alignment between datasets, or between the task setting itself. However, we found challenges in running this experiment as we describe below.
>
> *Using DROID for zero-shot evaluation on a task (i.e. no target / in-domain data) is an open problem, as is pointed out by the authors themselves in their paper [4].* There is no publicly-available zero-shot evaluation of visuomotor policies using this dataset either to the best of our knowledge. Nevertheless, we did try evaluating policies trained on different retrieved datasets, on 2 tasks in our study, and we did not achieve any success, as is summarized in table below.
>
> | Task | retrieve campose | retrieve spatial |
> |-|:-:|:-:|
> | *snack* | 0 | 0 |
> | *baking* | 0 | 0 |
>
> Hence, our empirical results show that models trained purely on retrieved datasets from DROID fail to perform a downstream task. We speculate that, while DROID is a good candidate dataset to enable a study such as ours, it currently does not have the required scale to enable zero-shot policy evaluation, making it impossible to study the efficacy of different retrieval strategies in a zero-shot setting as you recommended. As such, we leave this question open for future studies to answer.
>
> References:
>
> [1] LIBERO: Benchmarking Knowledge Transfer for Lifelong Robot Learning, Liu et al., 2023
>
> [2] BAKU: An Efficient Transformer for Multi-Task Policy Learning, Haldar et al., 2024
>
> [3] QueST: Self-Supervised Skill Abstractions for Learning Continuous Control, Mete et al., 2024
>
> [4] DROID: A Large-Scale In-The-Wild Robot Manipulation Dataset, Khazatsky et al., 2024

---

> ### Author Response · Authors · 2024-12-02
>
> We thank the reviewer for their valuable time spent in reviewing our work, and raising thoughtful questions which we addressed in our rebuttal. As the discussion period comes to an end, we sincerely request the reviewer to let us know if there are any more pending questions from their end, and adjust their score to reflect their final rating of our work. Thank you!

---

### Official Review · Reviewer_J7tN · 2024-11-02

**Soundness:** 2
**Presentation:** 3
**Contribution:** 3
**Rating:** 6
**Confidence:** 4

**Summary:**

This paper presents a systematic study of what factors matter most in large-scale robotics datasets for manipulation tasks. The authors introduce MimicLabs, a framework for generating controlled datasets with various dimensions of variation (DVs), and analyze dataset composition from both collector and retriever perspectives. They identify camera poses and spatial arrangements as crucial factors while finding object textures less important. The study's insights are validated on real robots and existing datasets like DROID.

**Strengths:**

The paper addresses the crucial question of dataset composition in large-scale robotic manipulation through a systematic analysis. The two-perspective approach (collector and retriever) offers practical insights for both dataset collection and utilization. The authors conduct comprehensive experiments and clear validation in both simulation and real-world settings. The definitions and methodology based on "DV" are novel.

**Weaknesses:**

1. Results heavily rely on MimicGen's simplified environments with basic textures and lighting. The conclusion that "texture alignment is less important" may not hold in real-world scenarios like those in DROID with complex lighting, shadows, and material properties (as in computer vision, lighting effects are often considered as a special kind of texture). Limited scene diversity compared to real datasets like DROID (8 simulated scenes vs. 52 real buildings)

2. Limited Policy Evaluation: Experiments primarily use Diffusion Policy, with only BC-RNN as comparison in MimicLabs. Performance gains might be attributed to Diffusion Policy's characteristics rather than dataset composition. Missing evaluation with modern approaches like VLA or transformer-based policies.

3. No clear metric for measuring skill similarity (e.g., is "picking a bowl" similar to "picking a cup"?) Lack of systematic handling of compound skills. Binary treatment of skill alignment without considering partial similarity.

4. Overlooked DV Interactions: No consideration of cases where a tejectory has conflicting values across different DVs (e.g., good camera pose but poor spatial arrangement). Missing analysis of trade-offs between different DVs when retrieving demonstrations. No principled approach for weighing the relative importance of different DVs when they conflict.

The limitations affect the generalizability and practical applicability of the paper's conclusions, particularly when scaling to more complex real-world scenarios or different learning algorithms.

**Questions:**

1. How do you handle cases where a demonstration might be valuable according to one DV but less useful according to another? Is there a principled way to make such trade-offs?
2. How do your findings generalize across different imitation learning approaches? Would VLA or other transformer-based policies show similar sensitivity to DVs?
3. How does your conclusion about texture alignment being less important translate to real-world scenarios with complex lighting conditions?
4. Given that similar objects often share manipulation strategies, why not consider 'picking a cup' and 'picking a bowl' as the same skill? Would grouping such similar skills during retrieval improve policy performance or potentially harm it?
5. Since DROID contains diverse task variations across 52 buildings while your simulation covers 8 scenes, could the limited scene diversity affect your DV importance rankings? Specifically, would more diverse scenes reveal additional important DVs?

---

> ### Author Response · Authors · 2024-11-23
> **Response to Reviewer J7tN (1/3)**
>
> We thank the reviewer for their time spent in going through our paper, giving insightful comments, and acknowledging that our work answers the crucial question of dataset composition in robotics. Please find below our response to their concerns, including *additional results* in the revised paper.
>
>
> > How do you handle cases where a demonstration might be valuable according to one DV but less useful according to another? Is there a principled way to make such trade-offs?
>
> Thank you for your insightful question. We agree that it would be interesting to see experiments that directly test how conflicting DVs affect target task success. For example, should we choose to diversify camera poses or spatial arrangements during data collection if only one of the two is possible?
>
> Our study provides some interesting conclusions even while analyzing one DV at a time. In the collector’s experiments, if we look at **Table 1** and **compare the relative increase in success rate** over the baseline dataset, we can infer what DV to prioritize in the case of conflict or when there is a fixed budget for data collection. Our results suggest that Cam Pose > Object Spatial > Table Texture > Object Texture. As validated in our results, we find these takeaways to hold in a retrieval setting, and even translated into real-world performance using retrieval from a large-scale real-robot dataset.
>
> Analyzing all possible combinations of DVs would vastly increase the number of experiments that need to be run. For example, determining whether to pick a demonstration that has aligned camera pose but misaligned spatial arrangements, or vice-versa, are just two of the N-factorial permutations of experiments that need to be run (where N is the number of DVs). Given that we analyze 5 DVs, this necessitates 120 different dataset compositions for one task and 1000s of high-end GPU hours for inference. While we do not have the resources to run each of these experiments, our data generation framework and method of analysis make it very easy to design such experiments. We also look forward to seeing these results if future researchers have the computational resources to directly answer this question.
>
>
> > How do your findings generalize across different imitation learning approaches? Would VLA or other transformer-based policies show similar sensitivity to DVs?
>
> Studying the effects of data compositions is computationally expensive due to multiple data re-collections, policy trainings and evaluations. Prior works that study dataset compositions for generalization in robotics, just evaluate one algorithm across multiple setups [1], and we agree that this common practice can introduce bias in the study. We evaluated BC-RNN and Diffusion Policy which are two very strong baselines considered in most current imitation learning literature [2]. They cover two very different architectures for processing history of observations (RNN in BC-RNN and CNN in Diffusion Policy) in addition to having different action heads and training objectives. Based on your suggestion, we have added yet another policy architecture, BC-Transformer, that processes observations using a transformer backbone. Hyperparameter details for this policy are added to **Table 6** in **Appendix F**. We ran our retrieval experiment on 5 tasks in the MimicLabs benchmark using **BC-Transformer**, and have included results in **Table 11** of **Appendix K**, also shown in the table below.
>
> | Task | #demos | Target only| obj/skill 	| +camPose 	| +objSpat | +recepSpat | +all | no obj/skill 	| +camPose 	| +objSpat | +recepSpat | +all |
> |-|:-:|:-:|:-:|:-:|:-:|:-:|:-:|:-:|:-:|:-:|:-:|:-:|
> |*bin carrot* | 10 | 53.33 | 90 | **93.33** | 83.33 | **93.33** | **93.33** | 50 | 43.33 | 53.33 | **73.33** | 40 |
> |*bin bowl* | 10 | 36.67 | 56.67 | 50 | 50 | **63.33** | 56.67 | **53.33** | **53.33** | 36.67 | 43.33 | 30 |
> |*clear table* | 20 | 30 | 40 | **53.33** | 33.33 | 23.33 | 43.33 | 13.33 | 23.33 | 23.33 | 23.33 | **33.33** |
> |*microwave teapot* | 20 | 10 | 13.33 | 13.33 | 20 | 16.67 | **30** | 16.67 | **23.33** | 6.67 | 3.33 | 10 |
> |*make coffee* | 50 | 6.67 | 10 | **20** | 10 | 16.67 | 13.33 | 10 | **20** | 3.33 | 3.33 | 13.33 |
>
> As we observe, our takeaways regarding the efficacy of different retrieval strategies from a diverse large-scale multi-task dataset still hold for a transformer-based policy. Finally, although VLA is becoming another popular policy for robot learning, in OpenVLA’s paper they report that “the final OpenVLA model is trained on a cluster of 64 A100 GPUs for 14 days, or a total of 21,500 A100-hours, using a batch size of 2048.” This is well beyond our computational budget for one training, much less the 100’s of policies we need to train to add OpenVLA as one of our baselines. We hope that future research can make use of our experimental framework to study dataset compositions for a larger variety of algorithms as they become more viable to use for the research community.

---

> ### Author Response · Authors · 2024-11-23
> **Response to Reviewer J7tN (2/3)**
>
> > How does your conclusion about texture alignment being less important translate to real-world scenarios with complex lighting conditions?
>
> Thank you for the insightful question. Our response is summarized as follows: (1) we highlight that our experimental takeaways regarding object textures from a collector’s perspective hold both in simulation and in real, (2) we acknowledge the case the reviewer pointed out and identify our general-purpose framework to be capable of providing meaningful conclusions for additional DVs of interest, and (3) we talk about the difficulty in quantifying lighting conditions in real-world scenarios discounting such a study in simulation.
>
> We found our conclusions about object textures to hold both in simulated and real world experiments, as well as in analyses from both the collector’s and retriever’s perspectives. In **Table 13** of **Appendix K**, we see that for all real-world tasks we achieve significant performance boost by simply adding more data without any heed to matching or even diverse object textures, and for 2 out of 3 tasks we obtain close to perfect performance. This is over and above our simulation results that showed that it was possible to alleviate disparity in object textures between target and co-training datasets through variation in other DVs (shown in **Table 1**).
>
> The experimental framework and simulation capabilities we provide in this study are *general-purpose*, and can enable future studies that further analyze the importance of DVs that researchers may uniquely identify. Using our framework, a downstream operator can identify if a convolution of lighting and textures creates a meaningful dimension of variation in their setup, and subsequently perform data collection uniquely tailored to their setting.
>
> We also want to point out that the choice of DVs in our study is guided by the currently available (and computable) meta-data in large-scale robotics datasets like DROID. Lighting conditions are hard to measure in the real world, and hence it was not meaningful to procedurally generate simulation data controlling this factor.
>
>
> > Given that similar objects often share manipulation strategies, why not consider 'picking a cup' and 'picking a bowl' as the same skill? Would grouping such similar skills during retrieval improve policy performance or potentially harm it?
>
> This is an interesting point - certainly measuring demonstration similarity is more difficult in some DVs than others. For skills, we chose to select similar demonstrations based on motion primitives (pick, place, push, pull) and target object. We believe this is an appropriate design choice for three primary reasons:
> 1. Different objects will indeed require different manipulation strategies. Even for the case of picking a bowl vs picking a cup, one might demonstrate picking a cup by its handle whereas a bowl by the rim. Overall, we believe it is difficult to pool manipulation skills beyond object categories.
> 2. Many other papers treat “pick X” as a different task than “pick Y” [3,4] so we are following a common trend in robot learning research.
> 3. Segmentation around motion primitives and object types is required for our data generation pipeline (MimicGen), so it is natural to retrieve skills in the same manner as this will yield closely matching demonstrations.
> Finding the correct metric of skill similarity is a non-trivial and open research problem. Determining similarity based on robot trajectories, object geometries, or assessment by a VLM could all be interesting avenues for future research. But we believe that our method of measuring skill similarity based on motion primitive and object type is a fairer assessment than just grouping all “picks” as the same skill.

---

> ### Author Response · Authors · 2024-11-23
> **Response to Reviewer J7tN (3/3)**
>
> > Since DROID contains diverse task variations across 52 buildings while your simulation covers 8 scenes, could the limited scene diversity affect your DV importance rankings? Specifically, would more diverse scenes reveal additional important DVs?
>
> Thank you for your question. While constructing new environments is inherently a time-intensive process, we would like to emphasize that our software framework is indeed capable of scaling to incorporate a broader range of environments. Nevertheless, we believe that 8 scenes is an appropriate scale for thoroughly testing and validating our hypotheses. In **Section 5.4**, we demonstrated that the insights gained from our experiments on the MimicLabs dataset do indeed transfer to actionable insights in the real world. Even though the DROID dataset has a greater amount of diversity, the same retrieval strategies used in our simulation environment lead to increased success rates on our real world tasks. That being said, we also provide the necessary tools and infrastructure for scaling up of the experimental setup, should future work seek to explore a larger variety of simulated environments.
>
> References:
>
> [1] Decomposing the Generalization Gap in Imitation Learning for Visual Robotic Manipulation, Xie et al., 2023
>
> [2] OpenVLA: An Open-Source Vision-Language-Action Model, Kim et al., 2024
>
> [3] LIBERO: Benchmarking Knowledge Transfer for Lifelong Robot Learning, Liu et al., 2023
>
> [4] RoboCasa: Large-Scale Simulation of Everyday Tasks for Generalist Robots, Nasiriany et al., 2024

---

### Official Review · Reviewer_Gs6e · 2024-11-03

**Soundness:** 4
**Presentation:** 2
**Contribution:** 3
**Rating:** 6
**Confidence:** 5

**Summary:**

This paper conducts a comprehensive empirical study on key factors in scaling robot manipulation datasets, offering valuable insights for both collectors and users.

**Strengths:**

1. This paper collects a large-scale robot manipulation dataset in simulation with diverse dimensions of variations to enhance understanding of each factor’s impact.

2. This paper provides practical suggestions on how to collect and leverage the data, and conducts real-world experiments on the DROID dataset to verify its effectiveness.

**Weaknesses:**

1.	In section 5.1, from the collector’s perspective, there’s only one task, “clear table”, which seems insufficient to draw conclusions (especially given that more tasks are actually used in the following sections).

2.	As illustrated in the paper, object geometry is vital to robot data. However, it is a pity that it is not included in the DVs of the experiments.

3.	Although the MimicLabs dataset generates different textures for the experiments, the textures are still primarily pure colors according to the appendix. More complex textures, such as plaid patterns or irregular textures, aren’t included.

4. There are a large number of experiments and various settings in this paper, but some of them are not explained clearly, which might be confusing (see the questions for details).

Overall, this paper conducts a nice trial to investigate the key factors in large-scale robot learning and provides many useful results and practical suggestions. However, more thorough experiments and more accurate explanations of experiments would make the conclusions more convincing.

**Questions:**

1.	How many tasks are included in the MimicLabs dataset? Different sections use various tasks from the dataset, but I am unsure how many tasks are there in total. According to the appendix, there seem to be five, but this is not shown in the main text or images.

2.	What exactly are the four co-training settings in Figure 2? There seems to be no exact definition of these settings, and different experiments actually use different numbers/types of co-training, making it harder to understand. It would be helpful to have a better description of all co-training settings.

3.	Why are all the co-training datasets in section 5.1 misaligned with the target? Can’t it contain the target distribution (as I saw in some experiments in other sections)?

4.	In section 5.3, the authors have an important finding that full retrieval might be less effective than other strategies using less data. However, a more common strategy used in other areas is to first pretrain on the full dataset and then fine-tune on the target dataset. I am wondering have the authors tried this setting?

---

> ### Author Response · Authors · 2024-11-23
> **Response to Reviewer Gs6e (1/3)**
>
> We thank the reviewer for their time spent in going through our paper, giving insightful comments, and acknowledging the valuable insights our study offers for data collection and retrieval in robotics. Please find below our response to their concerns.
>
>
> > In section 5.1, from the collector’s perspective, there’s only one task, “clear table”, which seems insufficient to draw conclusions (especially given that more tasks are actually used in the following sections).
>
> Thank you for raising this important point. Our response is organized as follows:
> 1. We provide further details about the motivation behind the experiments from the collector's perspective, highlight the multi-faceted nature of the single domain considered in simulation (Section 5.1), and subsequent extensive evaluation on a real robot (**Table 13** in Appendix K).
> 2.  We run additional experiments on a second task (`make coffee`) and show the same conclusions hold (see **Table 9** in Appendix K for additional results).
>
> The experiments from the **collector's perspective** offer a way for data collectors to test the importance of each dimension of variation (DV) to inform where they should allocate data collection effort. This is a large multi-faceted experiment where we collect multiple datasets with exhaustive variations along 4 DVs of the target task (camera poses, object textures, table textures, and spatial arrangements of the object) as well as 4 different dataset variations for co-training. We believe this framework and our in-depth analysis of a meaningful long-horizon task is valuable for future data collectors. In addition to **Table 1**, we have more results from the collector’s perspective in Appendix K where we analyze the motion primitive DV for data collection in this task (**Table 10**).
> This experimental design was specifically chosen to be adaptable to real-world settings with minimal data collection effort. Hence we also conducted this experiment in the **real world** (summarized in Section 5.4.1 and **Table 13**) on **3 different tasks**, collected over **1200 teleoperated demonstrations** and evaluated **39 different models across 13 different target-cotraining dataset combinations.**
> To further support our claims, we have added another challenging task in simulation to this analysis, `make coffee` (put the coffee pod in the coffee machine and close the lid). We find this task to be meaningfully different from the `clear table` task, with significantly different geometries, requiring higher precision for pick and place, and even different articulation. We present the results below, and have added them to **Table 9** in Appendix K of the revised paper.
>
> | Target task variation &#8595; | target only	|baseline| camPose | objTex |tableTex |objSpat |
> |-|:-:|:-:|:-:|:-:|:-:|:-:|
> | camPose 		|10	|16.67	|**43.33**	|3.33		|3.33	|16.67	|
> | objTex 		|10	|6.67	|6.67		|**36.67**	|**36.67**|**43.33**|
> | tableTex 		|10	|16.67	|10	 	|26.67		|**30**	|**33.33**|
> | objSpat		|16.67	|10	|10	 	|10		|20	|**26.67**|
>
> Similar to the setting in Table 1, different columns represent different co-training distributions with high variation along one DV, stemming from the baseline variation. The target task variations (rows) also contain misalignment along one DV from the baseline as well as the corresponding high-variation co-training dataset. We note that many of our takeaways are still the same in this experiment, showing (1) misaligned camera poses between target and co-training can hurt skill transfer, (2) disparity in object textures between target and co-training can be mitigated by other DVs, and (3) both camera poses and spatial arrangements are crucial aspects in data collection.
>
> > As illustrated in the paper, object geometry is vital to robot data. However, it is a pity that it is not included in the DVs of the experiments.
>
> We agree with the reviewer that object geometry is vital metadata in a demonstration that both a data collector and retriever should care about for efficient visuomotor policy learning. We want to re-emphasize that one of our main goals is to provide a conceptual methodology and software framework to conduct a data composition study, and we do not aim to be exhaustive due to the practical limitations of computation power and existing technology. For example, we rely on MimicGen for supersizing a set of human demonstrations, but generalizing this framework to significantly new object geometries is a research problem itself. We also point out that our analysis on the MimicLabs benchmark does retrieve along object geometry (and skill) to show the importance of such retrieval from a large-scale robotics dataset.

---

> ### Author Response · Authors · 2024-11-23
> **Response to Reviewer Gs6e (2/3)**
>
> > Although the MimicLabs dataset generates different textures for the experiments, the textures are still primarily pure colors according to the appendix. More complex textures, such as plaid patterns or irregular textures, aren’t included.
>
> We have included additional illustrations showing procedural texture generation and example plaid and irregular textures available in MimicLabs (see **Figure 7** in Appendix D.1 of the revised paper). Fractal textures can be tuned to look more diverse, and are a popular choice for texture generation in video games, among other applications. Specifically, our framework allows a user to provide how *grainy* and *turbulent* the generated textures should be, examples of which we show in the revised **Figure 7a**. MimicLabs also has the ability to load textures from a pool of files at each environment reset, with some examples including plaid and irregular patterns shown in **Figure 7c**. We will open-source our code upon publication which will allow users to create new scenes and tasks that leverage the entire pool of textures available in this benchmark beyond the testing we perform in this paper.
>
> > How many tasks are included in the MimicLabs dataset? Different sections use various tasks from the dataset, but I am unsure how many tasks are there in total. According to the appendix, there seem to be five, but this is not shown in the main text or images.
>
> Thank you for pointing this out. We have added explicit details about tasks in the MimicLabs dataset in **Appendix G.1** of the revised paper. MimicLabs contains over 3000 task instances encompassing different motion primitives for solving different tasks, varying camera poses, objects, table textures, spatial arrangements, and background scenes. We summarize the tasks included in the MimicLabs benchmark below:
> 1. `pick` X and `place` it in the bin (7 instances per lab)
> 2. `open` Y (2 instances per lab)
> 3. `close` Y (2 instances per lab)
> 4. `open` X, `pick` Y and `place` it in X (14 instances per lab)
> 5. `pick` X, `place` it in Y and `close` Y (14 instances per lab)
> 6. `turn on` stove
> 7. `turn off` stove
> 8. make coffee
>
> where X can be replaced by one of 7 distinct objects available in each lab for data collection and policy evaluation, Y can be a drawer or a microwave, distinct instances of which are available in each lab. In total, there are around 290 unique task instances in each lab, with skill-level overlap designed to test positive retrieval strategies. Additionally, for each task instance, we created multiple variations in camera poses (5), object and spatial arrangements (around 90 combinations), which create around 450 task instances in each lab, totaling to over 3000 instances across 8 labs. We have added this breakdown of tasks in **Appendix G.1** of the revised paper.
> The tasks in the main retrieval experiment are a single choice of the camera pose, object and receptacle spatial arrangment, out of the multiple available in the benchmark (see **Table 8** in Appendix G.1). We also choose them to be in increasing order of hardness, as listed below:
> 1. Bin carrot: single-mode grasping strategy
> 2. Bin bowl: multi-modal grasping strategy
> 3. Clear table: pull drawer and pick-place a bowl
> 4. Microwave teapot: pick-place a mug and push microwave door
> 5. Make coffee: high precision, receptacle not available in any other lab for retrieval
>
> > What exactly are the four co-training settings in Figure 2? There seems to be no exact definition of these settings, and different experiments actually use different numbers/types of co-training, making it harder to understand. It would be helpful to have a better description of all co-training settings.
>
> Thank you for pointing out that additional details could be beneficial. We have added a discussion around the four considered cases of distributions in Appendix B of the paper, which we also summarize below.
>
> Figure 2 shows the following 4 cases, starting from top-left going in clockwise order:
> 1. not-diverse and misaligned
> 2. diverse and misaligned
> 3. diverse and aligned
> 4. not-diverse and aligned (perfect alignment)
>
> As shown in our experiments (Section 5.1), cases 1 and 2 (diverse or not, with misalignment) are important to study from a collector's perspective where the goal is to find out which DV the collector should add diversity in given the **worst-possible scenario** i.e. misalignment. Cases 3 and 4 are important from a retriever's perspective (Section 5.2) given the assumption that large-scale robotics datasets contain high diversity along all DVs. Subsequently, by retrieving datasets from the entire MimicLabs dataset (Section 5.3), our study seeks to understand if retrieving datasets with perfect alignment (case 4) between target and co-training datasets is useful in a practical situation for boosting performance in the target domain.

---

> ### Author Response · Authors · 2024-11-23
> **Response to Reviewer Gs6e (3/3)**
>
> > Why are all the co-training datasets in section 5.1 misaligned with the target? Can’t it contain the target distribution (as I saw in some experiments in other sections)?
>
> Thank you for raising this point for clarification. The goal of the collector’s experiment is to build an understanding about which DV the collector should add diversity in given the **worst-possible scenario**, i.e. when there is guaranteed misalignment between the target and co-training datasets, which is why we consider misaligned target and co-training datasets for this analysis. To further extrapolate this point, if the target distribution’s support was a subset of that of the co-training distribution, the downstream operator might want to perfectly align them, which is a case we study from a retriever’s perspective where such a scenario exists along all DVs. It is also important to note that in the collector’s experiment, only one DV is misaligned between target and co-training distributions. This for two reasons, (1) it tells us which DVs are **necessary** to add diversity in, and (2) it creates an easy framework to find DVs that are crucial for data collection in a real-robot experiment through minimal data collection effort.
>
>
> > In section 5.3, the authors have an important finding that full retrieval might be less effective than other strategies using less data. However, a more common strategy used in other areas is to first pretrain on the full dataset and then fine-tune on the target dataset. I am wondering have the authors tried this setting?
>
> Thank you for raising this interesting point. Many existing research works in imitation learning have adopted the paradigm of co-training when using large-scale datasets [1,2,3] as it provides a stable training methodology without the need for fine-grained tuning of optimization parameters or neural architectures, choosing which parts of the model to freeze/train, etc. so as to prevent overfitting to in-domain data. The focus of our study was to understand how visuomotor policy learning can be enhanced by using large-scale datasets, where multiple datasets are used to train a single policy, and the policy is evaluated on the downstream task for which an operator collects minimal data. Owing to the large-scale adoption of a co-training paradigm, we chose to extensively evaluate our data collection and retrieval strategies in this setting.
>
> Nonetheless, based on your suggestion we now include pretraining+finetuning results in the MimicLabs benchmark on 3 tasks using a BC-RNN policy. Interestingly, we see similar takeaways in this setting using our retrieval strategies as we saw when co-training. Pre-training with the retrieved datasets achieves significantly higher performance compared to when training from scratch on just in-domain data, with our proposed retrieval strategies always providing a performance boost.
>
> | Task | target only 	| obj/skill 	| +camPose 	| +objSpat 	| +recepSpat 	| +all 	|
> |-|:-:|:-:|:-:|:-:|:-:|:-:|
> | *bin carrot* 	| 50    	| 76.67 	| 80  		|83.33		|**86.67**	| 83.33	|
> | *bin bowl* 	| 33.33 |  76.67	| 73.33		|66.67		|66.67		|**83.33**|
> | *clear table* 	| 36.67	| 23.33	 	|**60**	 	| 20		| 20		|46.67	|
>
> | Task 		| no obj/skill 	| +camPose 	| +objSpat 	| +recepSpat 	| +all 	|
> |-|:-:|:-:|:-:|:-:|:-:|
> | *bin carrot* 	|60		|73.33		|**80**		|63.33		|76.67	|
> | *bin bowl* 	|60		|66.67		|66.67		|**73.33**	|60	|
> | *clear table* 	|43.33	 	|30	 	|26.67		|46.67		|**50**	|
>
> We have added these results in **Table 12** of **Appendix K** in the revised paper. We hope these additional results will alleviate the concerns about the wide-range applicability of our results.
>
> References:
>
> [1] DROID: A Large-Scale In-The-Wild Robot Manipulation Dataset, Khazatsky et al., 2024
>
> [2] Open X-Embodiment: Robotic Learning Datasets and RT-X Models, Open X-Embodiment Collaboration, 2023
>
> [3] Re-Mix: Optimizing Data Mixtures for Large Scale Imitation Learning, Hejna et al., 2024

---

> ### Author Response · Authors · 2024-12-02
>
> We thank the reviewer for their valuable time spent in reviewing our work, and raising thoughtful questions which we addressed in our rebuttal. As the discussion period comes to an end, we sincerely request the reviewer to let us know if there are any more pending questions from their end, and adjust their score to reflect their final rating of our work. Thank you!

---

> ### Comment · Reviewer_Gs6e · 2024-12-02
>
> Thanks for the detailed response during the rebuttal period. My concerns have been addressed, so I decide to improve the score to 6. Please include the new results into the final version.

---

### Author Response · Authors · 2024-11-23
**Response to All Reviewers**

We thank the reviewers for their time and effort and greatly appreciate their positive comments and insightful questions about our work.

We are delighted that reviewers found our study to be comprehensive and offer practical suggestions (Gs6e), our paper to address the crucial question of dataset composition (J7tn), our experiments to be extensive (D5QM), and our approach for testing different dataset compositions to be novel and impactful (Cfcj).

The reviewers raised a number insightful comments and questions, which we thoroughly addressed in our individual responses. Moreover, we also provided additional results, which we summarize below:

1. **Additional collector experiment** on a simulated task (`make coffee`) (**Table 9** in **Appendix K**)

2. Evaluating our retrieval strategies on the entire MimicLabs benchmark using a **transformer-based behavior cloning policy** (**Table 11** in **Appendix K**)

3. Showing the efficacy of our retrieval strategies in a **pre-training followed by fine-tuning setup**, prove further applicability of our results and evaluation framework (**Table 12** in **Appendix K**)

We have revised our paper based on feedback from reviewers (changes highlighted in blue), with additional results in the appendix as above. We sincerely hope that the reviewers will consider our rebuttal response to their concerns and additional experiments towards their final rating of our paper!

---

### Meta-Review · Area_Chair_xwnY · 2024-12-24

**Metareview:**

This paper presents a comprehensive study on key factors that influence the effectiveness of large-scale robotics datasets for manipulation tasks. The authors introduce MimicLabs, a framework to generate synthetic datasets with controlled variations, which allows for a systematic analysis of dataset diversity. Through detailed experiments, the study highlights camera poses and spatial arrangements as being critical for dataset composition, while object textures seem to have comparatively less impact. These findings are validated across simulations, some real-world tasks, and datasets like DROID, showing up to 70% improvement in downstream policy performance in certain scenarios.

Almost unanimously, the reviewers appreciated the thorough empirical study on scaling robot manipulation datasets, the introduction of the MimicLabs framework with its controlled dimensions of variation (DVs), and the focus on key factors like camera poses and spatial arrangements for dataset composition. They also noted that the paper tackles an important and practical problem in the design of robotics datasets. Reviewers raised questions about single-task analysis from the collector's perspective, the choice of co-training versus other paradigms, multiple DVs, the role of texture, and data diversity. Through additional collector experiments, more base policy choices, and pretraining followed by fine-tuning results, many of these concerns were partially resolved.

The AC concurs with the unanimous recommendation from the committee to accept this paper. The paper raises an important and novel research question and convincingly provides insights that are valuable for the robot learning community.

**Additional Comments On Reviewer Discussion:**

1. Reviewer `Gs6e` increased their score from 5 to 6, noting that their concerns had been adequately addressed.
2. Reviewer `J7tN` raised their score from 5 to 6. While concerns about multiple DVs and the hard categorization of skills remained after two rounds of discussion, they remarked that "the paper's current contributions meet the threshold for acceptance."
3. The AC finds that the authors have convincingly addressed the concerns raised by Reviewer `D5QM`, who initially rated the paper a 6.
4. Reviewer `CfcJ` acknowledged the rebuttal but maintained their score at 6.

In post-rebuttal discussions with the committee, it was noted that despite the authors' efforts, some weaknesses identified earlier still persist. Nonetheless, the research direction exploring what matters in learning from large-scale datasets for robot manipulation is seen as both interesting and valuable. The authors' additional thoughts on this topic were appreciated.

---

### Decision · Program_Chairs · 2025-01-22

Accept (Poster)